# Streamflow variability over 1881-2011 period in northern Quebec: comparison of hydrological reconstructions based on tree rings and on geopotential height field reanalysis

P. Brigode[1,2,*], F. Brissette[1], A. Nicault[3], L. Perreault[4], A. Kuentz[5], T. Mathevet[6], and J. Gailhard[6]

[1]Ecole de technologie supérieure de Montréal, Montréal, Canada
[2]Ouranos, Montréal, Canada
[3]ECCOREV, Aix-en-Provence, France
[4]IREQ, Varennes, Canada
[5]SMHI, Norrköping, Sweden
[6]DTG, DMM, Electricité de France, Grenoble, France
[*]Now in: Université Côte d'Azur, CNRS, OCA, IRD, Géoazur

*Correspondence to:* Pierre Brigode (pierre.brigode@unice.fr)

**Abstract.** Over the last decades, different methods have been used by hydrologists to extend observed hydro-climatic time series, based on other data sources, such as tree rings or sedimentological datasets. For example, tree ring multi-proxies have been studied for the Caniapiscau Reservoir in northern Quebec (Canada), leading to the reconstruction of flow time series for the last 150 years. In this paper, we applied a new hydro-climatic reconstruction method on the Caniapiscau Reservoir and

compare the obtained streamflow time series against time series derived from dendrohydrology by other authors on the same catchment and study the natural streamflow variability over the 1881-2011 period in that region. This new reconstruction is based, not on natural proxies, but on a historical reanalysis of global geopotential height fields, and aims firstly to produce daily climatic time series, which are then used as inputs to a rainfall-runoff model in order to obtain daily streamflow time series. The performances of the hydro-climatic reconstruction were quantified over the observed period, and showed good performances,

both in terms of monthly regimes and interannual variability. The streamflow reconstructions were then compared to two different reconstructions performed on the same catchment by using tree ring data series, one being focused on mean annual flows, and the other one on spring floods. In terms of mean annual flows, the interannual variability of the reconstructed flows were similar (except for the 1930-1940 decade), with noteworthy changes seen in wetter and drier years. For spring floods, the reconstructed interannual variabilities were quite similar for the 1955-2011 period, but strongly different between 1880 and

1940. The results emphasize the need to apply different reconstruction methods on the same catchments. Indeed, comparisons such as those above highlight potential differences between available reconstructions, and finally, allow a retrospective analysis of the proposed reconstructions of past hydro-climatological variabilities.

# 1 Introduction

## 1.1 Challenge of decadal hydrological variability

Time series of streamflow observations, which constitute the basis for all hydrological analyses, are generally characterized by a relatively short record period, typically ranging from several years to several decades. In fact, the average length of 6945 daily streamflow series collected by the Global Runoff Data Center, and available worldwide, is 44 years (GRDC, 2015). The information extracted by hydrologists from these time series (in the form of statistical indices, calibration of model parameters, etc.) are generally used for water resource management, for instance for hydropower generation mid- to long-term planning. The short record period is a major issue for hydrologists since it may be insufficient to capture and provide a clear understanding of the decadal variability of hydrological processes. For example, after studying a 90-year long daily streamflow series of the Po River (Italy), and highlighting significant natural variability at the decadal scale, Montanari (2012) stated that "more research efforts are needed to improve the interpretation of such long-term fluctuations". Studying natural variability requires long instrumental records (typically longer than 100 years), but such long time series are non-existent in remote regions such as northern Quebec (Canada). The length (number of years) of 221 observed streamflow time series from Quebec - extracted from the cQ2 (*Impact des Changements Climatiques sur l'hydrologie (Q) au Québec*) database Guay et al. (2015) - is shown in Fig. 1b and c, highlighting that very few series have more than 50 years of data. Hydrological decadal variability is crucial in this region, since it is home to some of the largest hydropower systems in the world; as well, significant inter-annual inflow variability has been recorded in several Quebec catchments (e.g. Perreault et al. (2000, 2007); Jandhyala et al. (2009)). The few decades of observations available for this region are not sufficient to allow a robust analysis of multi-decadal hydrological variability, and thus, raise the issue of the reconstruction of past hydrology, i.e., occurring before the systematic recording of streamflows.

## 1.2 Reconstruction of past hydrology

Over the past decades, different methods have been used by hydrologists to reconstruct natural flows on catchments of interest, depending on available data. These methods may be classified into two groups, according to the temporal resolution of the reconstructed series.

The first group brings together the methods based on long and continuous hydro-climatic series constructed with daily or sub-daily observations, and consequently, allowing the reconstruction of streamflow time series at a fine temporal scale (e.g., daily resolution). When long streamflow series are available for other catchments close to the one under study, classical statistical regressions or other regionalization methods could be applied for the reconstruction (e.g., Hirsch, 1982; Hernández-Henríquez et al., 2010; Arsenault and Brissette, 2014). The paired catchment approach - consisting of calibrating and then using a streamflow-streamflow model - could also be used (e.g., Andréassian et al., 2012). When long climatic series (typically covering precipitation and temperature) are available in the studied region, the reconstruction could be done by using a rainfall-runoff model, in order to transform the climatic series into streamflow series (e.g., simulation of 124 years of streamflow for the Thames River (UK) by Crooks and Kay (2015)).

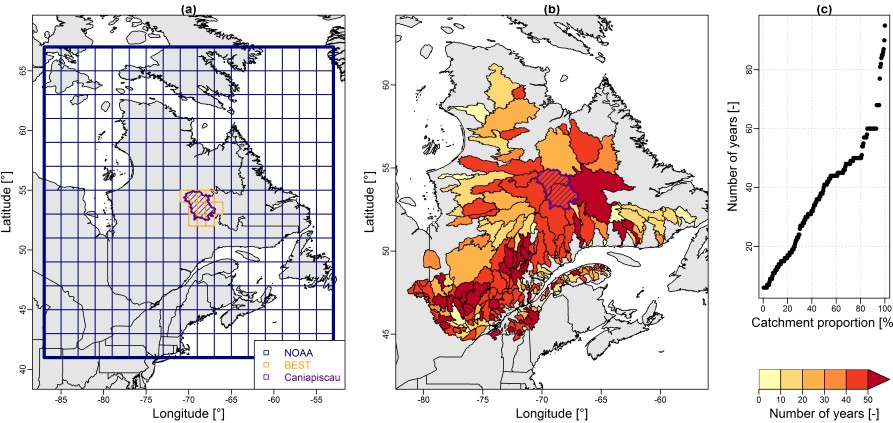

**Figure 1.** (a) Datasets used for the hydro-climatic reconstruction: the extension of the 20CR grid points used is shown in blue, while the BEST grid points used are highlighted in purple. The Caniapiscau reservoir catchment is plotted in purple. (b) Spatial distribution and (c) distribution of the length (number of years) of the observed streamflow series for 211 catchments in Quebec, extracted from the (cQ)2 database, Guay et al. (2015).

The second method is based on continuous or discrete series of paleo-indicators, generally producing reconstructed series at seasonal or annual resolutions (Bradley, 1999). The most natural proxies used for hydrological reconstructions are sediment stratigraphy (e.g., Thorndycraft et al., 2005) and tree ring series (see reviews by Loaiciga et al. (1993); Meko and Woodhouse (2011)). This latter proxy for streamflow reconstruction, referenced as dendrohydrology (Loaiciga et al., 1993), is analyzed in
a bid to reconstruct past hydro-climatological variations of a given catchment by studying tree ring width variations among different trees sampled in the same region. Reconstructed streamflow series are obtained by applying either direct or indirect methods. The direct methods aim to link tree ring series with streamflow series through statistical models calibrated over an observation period (e.g., in Tasmania (Australia) by Allen et al. (2015) and in the southeastern United States by Patskoski et al. (2015)). The indirect methods aim firstly to reconstruct climatic series, such as temperature or precipitation, and secondly,
to transform these climatic series into streamflow series through rainfall-runoff models (e.g., in the Western US by Gray and McCabe (2010); Saito et al. (2015)). These methods allow the continuous reconstruction of the annual or seasonal water balance of a given region, over long time periods. Additionally, other information could be extracted following tree ring analysis and used to reconstruct discrete chronologies of extreme hydrological events. For example, George and Nielsen (2003) used anatomical tree ring signatures to reconstruct paleofloods of the Red River in Manitoba (Canada).
Recently, dendrohydrological methods have been successfully applied in boreal environments, characterized by a rarity of long hydro-climatological series. For example, Nicault et al. (2014) used tree ring multi-proxies (tree ring widths, tree ring densities and tree ring stable isotope ratios) to produce spring, summer and annual flow series of the Caniapiscau Reservoir in northern Quebec (Canada) for the 1800-2000 period. On the same catchment, Boucher et al. (2011) used both continuous series (tree ring minimal density measurements) and discrete series (with ice-scars due to ice abrasion during floods) to produce

spring flood series for the 1850-1980 period. These two reconstructions revealed significant flow variability in this region, both in terms of annual flows and flood frequency. It should be noted that the Caniapiscau Reservoir is the most upstream and one of the largest reservoirs of the La Grande complex, which is one of the biggest hydro-power generation complexes in the world, with a total installed generating capacity of 17,418 megawatts. Decadal hydro-climatological variability in this region thus provides important information concerning the long-term planning of hydro-power generation.

## 1.3  Scope of paper

Although the above-mentioned hydrological reconstructions were associated with good verification statistics on the calibration period, the lack of observed streamflow data did not allow a rigorous independent verification of those reconstructions. An alternative solution involved carrying out new reconstructions based on different proxies and different methods, and then, as an additional verification step, analyzing the consistency between the different reconstructions. Comparisons of streamflow reconstruction methods are rare in the literature, and the Caniapiscau Reservoir catchment offers an interesting case study since various tree ring reconstructions have been performed there. Thus, our objective is to apply a new reconstruction method on the Caniapiscau Reservoir, in order to compare the obtained streamflow series with series obtained by dendrohydrology and to study the observed streamflow variability over the 1881-2011 period. This new reconstruction is based, not on natural proxies, but on a historical reanalysis of geopotential height fields. A climatic ensemble was reconstructed at the daily resolution using the ANATEM methodology (Kuentz et al., 2015), a resampling method based on synoptic situation similarities between days (found by looking at the geopotential height reanalysis), with a sampling of observed climatic series for a given time period (*the observation period*) over a longer time period (*the reconstruction period*). Then, a rainfall-runoff model - previously calibrated on the observed period - was used to transform this climatic ensemble into a streamflow ensemble. The performances of the hydro-climatic reconstructions and of the rainfall-runoff model calibration were firstly evaluated over the observed period, by comparing the reconstructions and the simulations with the observations. Secondly, the tree ring based on the ANATEM centennial reconstructions were compared, and finally, the long-term hydrological variability of the Caniapiscau Reservoir was discussed.

## 2  Data

### 2.1  Datasets used for the climatic reconstructions

#### 2.1.1  Geopotential height reanalysis

The climatic reconstruction method applied in this study (fully detailed in the following section) is based on finding similarity between days at the synoptic scale. The similarity is based on geopotential height fields over a given spatial domain. A geopotential height is the height above sea level of a given pressure level. Note that for pressure levels close to sea level (typically 1000 hPa), the geopotential height can sometimes be negative. The analysis of geopotential height fields over a given domain describes the spatial distribution of high/low pressure systems upon which similarity in between days can be measured. Sev-

eral long-term geopotential height reanalysis have been produced during the last decade, in order to study climate variability and climate change over the last century. The geopotential height reanalysis used in this study was drawn from the 20th Century Reanalysis V2c data, provided by the NOAA/OAR/ESRL PSD, Boulder, Colorado, USA, available from their Web site at http://www.esrl.noaa.gov/psd/ (Compo et al., 2011). This global reanalysis (hereafter denoted as 20CR), assimilating only

surface observations of synoptic pressure, monthly sea surface temperature, and sea ice distribution, spans the period of 1851 to 2011, with a six-hourly temporal resolution and a 2° spatial resolution. For each day, two levels were considered here, 1000 hPa at 0h and 500 hPa at 0h. The geopotential height fields were extracted over an area covering the entire province of Quebec, with 221 grid points, as shown in Fig. 1a. Of the 56 ensemble members constituting the 20CR reanalysis, the members 1 to 5 were extracted and used over this region (see section 3.3 for more details).

**2.1.2   The quest for centennial climatic series in northern Canada**

Centennial and continuous climatic series are rare in Canada, and almost non-existent in remote high-latitude regions, such as northern Quebec (Cowtan and Way, 2014). In this study, there is a need for both consistent and very long (> 100 years) climatic series. Mekis and Vincent (2011) and Vincent et al. (2012) built two databases of "adjusted and homogenized" air temperature and precipitation series, respectively, both available at monthly and daily resolutions for all of Canada. These databases were

specifically created for use as references in climate change impact studies. During their creation, care was taken to correct any errors that may surface, and to account for any shifts that may occur as a result of stations being moved or of changes in measurement instruments that may be present in the climatic series observed. Nevertheless, the average length of such series in northern Quebec is 50 years, which is considered too short for this work or for any study concerning natural climatic variability.

In Quebec, the few long climatic series (> 100 years) available are generally for large cities, which are all located in the

southern part of the province. These series are rarely continuous at the daily time scale, and are derived from different sources; as a result, producing good quality continuous series therefore requires a lot of work. For example, Slonosky (2014) compiled data from numerous sources (mainly from the cities of Québec and Montreal) to produce continuous daily temperature series for the St. Lawrence Valley region for the 1798-2010 period. In northeastern Canada, two sources of such historical data exist. First, the Moravian missionaries, who have been living among the Inuit in the Labrador coastal region since 1771, have

measured and recorded climatic variables (Demarée et al., 2010). Secondly, interesting qualitative information for the Hudson Bay and the James Bay (northwestern Quebec) 19th century climate are present in the Hudson's Bay Company trade post journals. Wilson (1988) compiled these data and produced summer temperature series and a wetness index for this region, and the series was then used by Bégin et al. (2015) as a reference series for comparisons with their climate reconstruction of the Canadian northeastern boreal forest. Unfortunately, no such data sources are present in the interior part of northern Quebec.

**2.1.3   A reanalysis as local reference temperature series**

For the air temperature, the Berkeley Earth Surface Temperature (hereafter denoted as BEST) analysis has been used, taken from the http://berkeleyearth.org/ Web site (Rohde et al., 2013). BEST is a gridded air temperature reanalysis for lands, starting in 1753 at the monthly resolution, and in 1880 at the daily resolution, with a 1° spatial resolution. A daily catchment series has

been assembled for the 1880-2011 period by averaging the 11 BEST grid points covering the Caniapiscau reservoir catchment, highlighted in Fig. 1. Note that this reanalysis was recently used in northeastern Canada by Way and Viau (2014), in their study of past air temperature variability in Labrador.

## 2.2 Caniapiscau reservoir catchment

In Quebec, 99% of the produced electricity comes from hydropower generation systems. The La Grande water resources system, located in northern Quebec and operated by Hydro-Québec (HQ), is one of the most important hydropower systems in the world, with an installed capacity of 17,418 megawatts (the Three Gorges Dam is the most important hydropower system in the world with a total installed capacity of around 22,000 megawatts). This system produces 50% of the total energy generated by HQ. The Caniapiscau hydroelectric reservoir catchment is the first dam of the La Grande operational chain (the Brisay power

plant installed at the outlet of the Caniapiscau reservoir is ranked as the 9[th] with an installed capacity of around 500 megawatts) and is a 37,328 km$^2$ snowmelt-dominated catchment. Figure 2 illustrates the hydro-climatic context of the Caniapiscau reservoir catchment. The catchment elevation (SRTM data, Jarvis et al. (2008)) ranges from around 500 to 900 m a.s.l., with the highest elevation areas located in the southern parts of the catchment. The daily streamflow series (a) and the monthly regimes (c) show the strong snow-dominated signature of the catchment, with an annual flood observed due to snowmelt during the month June.

On average, the mean annual precipitation and runoff are around 800 mm (with around 300 mm falling as snow) and 650 mm, respectively, on the Caniapiscau reservoir, and the mean annual temperature is around -3.6°C. Catchment climatic data used in this study consists of daily series of minimum, mean and maximum air temperature and of total precipitation, available for the 1950 to 2011 period. This dataset was produced by HQ, using kriging methods (Tapsoba et al., 2005). Daily streamflow series are available from 1962 to 2011. Note that only the 1962-1979 period was considered for the rainfall-runoff model calibration

here, since the Caniapiscau Dam was built during the 1980-1982 period, and streamflow series available for 1982 to 2011 are naturalized flows produced by HQ. Nevertheless, this second period (1982-2011, mean annual values are plotted in grey in Fig. 2b) will be used as a validation period for the reconstruction.

## 2.3 Reconstructed yearly streamflow series from tree rings

Two yearly time series of Caniapiscau Reservoir flows have been used here for comparison at the centennial scale: (i) the series

of annual flows proposed by Nicault et al. (2014) for the 1800-2000 period, and (ii) the series of spring floods proposed by Boucher et al. (2011) for the 1850-1980 period. The first yearly time series was processed from continuous tree ring series derived from 20 black spruce (*Picea mariana* [Mill.] BSP) sites located within 200 km around the Caniapiscau reservoir. Two reconstruction methods were used (Partial Least Square regression (PLS) and Best Analogue Methods), and the reconstructions obtained were combined in a single composite reconstruction. The second yearly time series was processed from ice-scar time

series derived from a small lake located next to the Caniapiscau reservoir and using tree ring densities obtained from 12 black spruce sites. A new transfer model technique based on Generalized Additive Model (GAM) theory was used to process spring flood reconstructions.

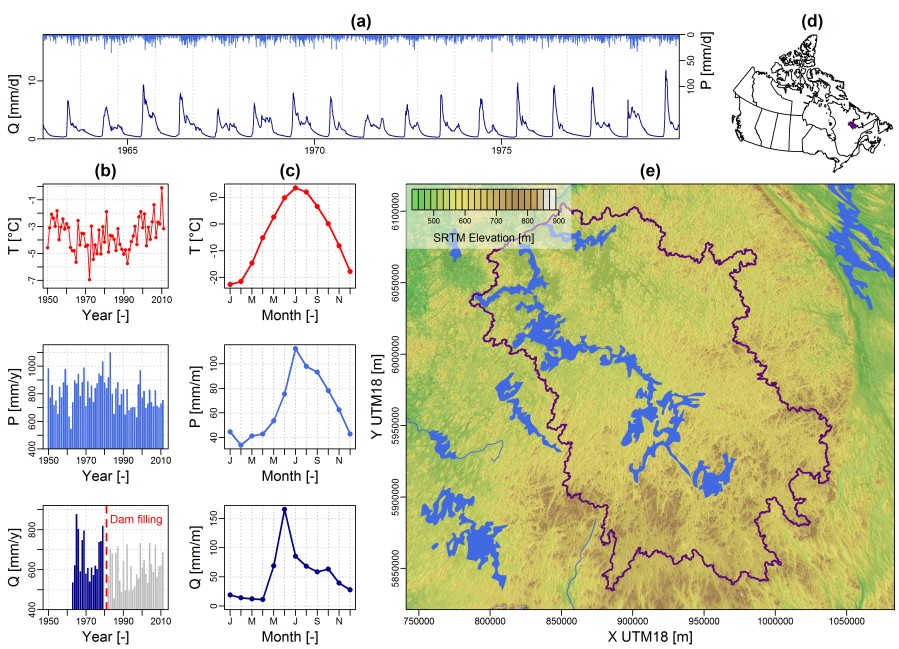

**Figure 2.** Hydro-climatic context of the Caniapiscau reservoir catchment: (a) observed daily streamflow and precipitation time series used for the rainfall-runoff model calibration (1962-1979), (b) temperature, precipitation and streamflow mean annual series, (c) temperature, precipitation and streamflow monthly regimes, (d) catchment location within Canada, and (e) SRTM elevation data. Monthly regimes were calculated for the 1950-2011 period for temperature and precipitation, while for the 1962-1979 period, the calculations were for streamflow.

## 3 Methodology

### 3.1 General streamflow reconstruction methodology

The general methodology consists in the reconstruction of an ensemble of daily climatic time series (with the ANATEM method) and of the transformation of this daily climatic ensemble into a daily streamflow ensemble, using a rainfall-runoff model. The ANATEM method (Kuentz et al., 2015) is built on the combination of two approaches: (i) the ANA (which stands for "ANAlogue") approach, that aims to find, for a given day, a given number of analogue days, based on the similarity of synoptic circulation (Obled et al., 2002; Schenk and Zorita, 2012) and (ii) the TEM (which stands for "TEMoin", the French word for "witness") approach, which is a basic regression model that uses a continuous and long-term reference (the witness) climatic series to reconstruct past climate. The ANATEM method thus allows the reconstruction of the climate of the past by combining synoptic information (ANA approach) with local climatic observations (TEM approach). Finally, this method allows the production of an ensemble of daily climatic time series by the selection of several analogues for any given day. For a complete description of the ANATEM method and an evaluation of its performance at the regional scale (French Alps), see Kuentz et al. (2015). The rainfall-runoff transformation is done here with GR4J (Perrin et al., 2003), a daily lumped

continuous rainfall-runoff model and its snowmelt routine, CemaNeige (Valéry et al., 2014a). GR4J and CemaNeige have 4 and 2 free parameters to calibrate, respectively, using the observed streamflow data available on the studied catchment. The whole streamflow reconstruction methodology - performed in the R-project environment (2014, http://www.r-project.org/) - is carried out in four steps (see Fig. 3):

– **Step 1: calibration of the rainfall-runoff (R-R) model.** The rainfall-runoff model is calibrated on the observed streamflow data.

     – **Step 2: finding analogue dates (ANA).** Synoptic states are compared in order to find analogue days for each day of the reconstruction period, amongst the days of the observation period.

     – **Step 3: reconstruction of a daily climatic (P and T) ensemble (ANATEM).** The best analogue obtained at step 2 are
10         stochastically resampled and long-term reference climatic series are used (if available) to improve the resampled series.

     – **Step 4: reconstruction of a daily streamflow ensemble.** The climatic ensemble is transformed into a streamflow ensemble using the rainfall-model parameter set obtained at step 1.

These four steps are further detailed hereafter.

## 3.2    Step 1: calibration of the rainfall-runoff model

The GR4J (Perrin et al., 2003) rainfall-runoff model was used to transform the climatic ensemble into ensembles of streamflow time series. GR4J is an efficient and parsimonious (only four free parameters to be calibrated) daily lumped and continuous model, which, when it is combined with its snow accumulation and melt routine, CemaNeige (Valéry et al., 2014a), is well suited for the hydrological modeling of snow-dominated catchments. GR4J and CemaNeige (model-pair hereafter denoted as CemaNeigeGR4J) were recently evaluated over several catchments located in Quebec (e.g., Seiller et al., 2012; Valéry et al.,
2014b) and dhowed good modelling performances. The structure of the CemaNeigeGR4J model is presented in Fig. 3. GR4J is based on two non-linear stores (production and routing stores) and a unit-hydrograph, while CemaNeige is a degree-day snow accounting routine, which divides the studied catchment into five elevation bands. CemaNeigeGR4J uses as inputs daily series of precipitation, minimal and maximal air temperatures and a daily potential evapotranspiration series, calculated using (Oudin et al., 2005) formula, designed for rainfall-runoff modelling. CemaNeigeGR4J produces daily streamflow series. GR4J and
CemaNeige have 4 and 2 free parameters to calibrate, respectively. These 6 parameters - highlighted in Fig. 3 and described in Table 1 - were calibrated conjointly over the same calibration period using a local gradient search procedure, applied in combination with pre-screening of the parameter space (Perrin et al., 2008). The Kling and Gupta Efficiency criterion (Gupta et al. (2009), hereafter denoted as KGE) was used as objective function. The KGE criterion ranges between $-\infty$ and 1 (perfect simulation) and is estimated as follows:

$$KGE = 1 - \sqrt{(\beta - 1)^2 + (\alpha - 1)^2 + (r - 1)^2} \qquad (1)$$

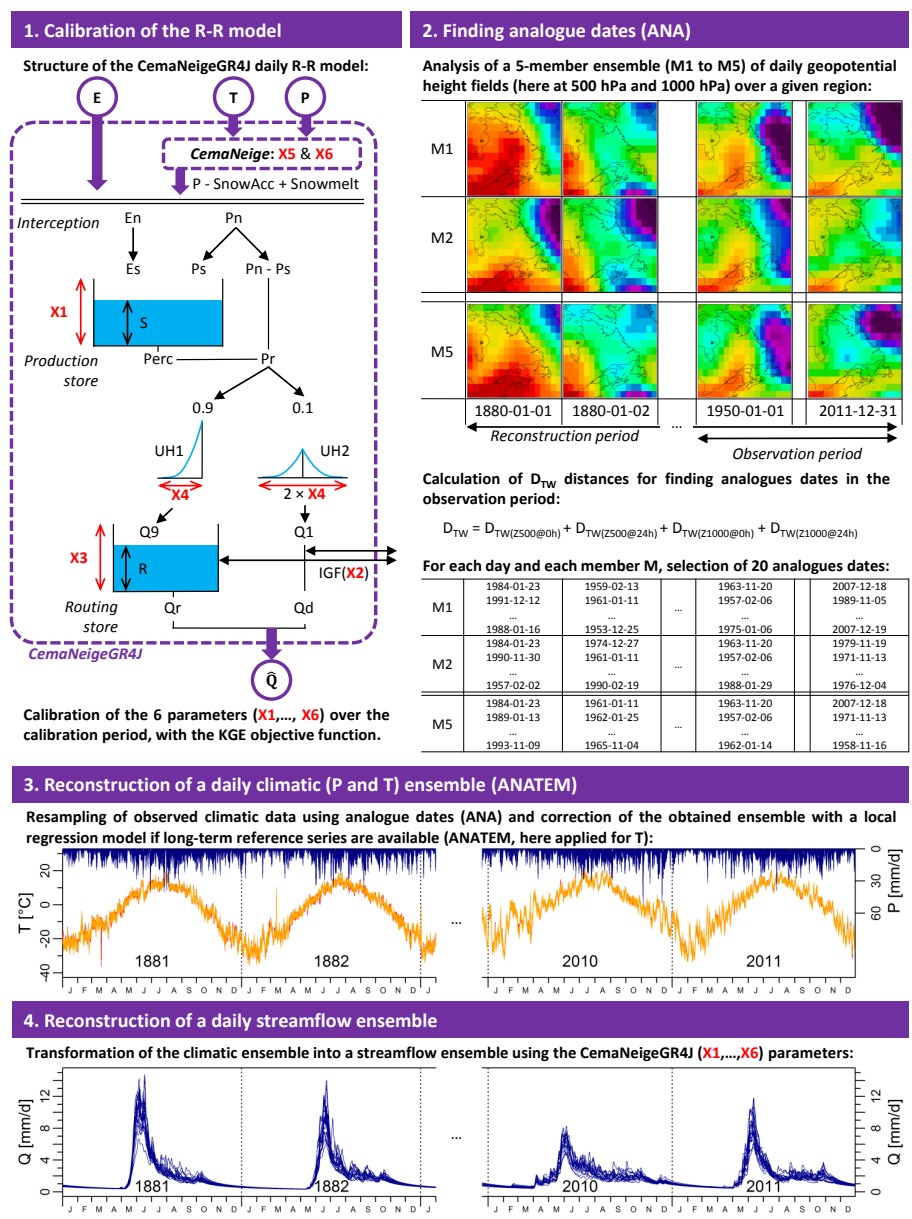

**Figure 3.** Illustration of the four-step methodology used for the reconstruction of a daily streamflow ensemble (R-R stands for rainfall-runoff, E for potential evapotranspiration, T for air temperature, P for precipitation, Q for streamflow).

With:

- $\beta$: ratio between the means of the simulated and observed streamflow time series; this quantifies the simulation bias, and ranges between 0 and $+\infty$ (values > 1 indicate a model overestimation).

**Table 1.** Description and final values of the 6 free parameters of the CemaNeigeGR4J model after being calibrated over the observed streamflow series of the Caniapiscau catchment.

| Parameter | Description (and unit) | Calibrated values |
|---|---|---|
| X1 (GR4J) | Capacity of the production store (mm) | 405 |
| X2 (GR4J) | Water exchange coefficient (mm/day) | 3.06 |
| X3 (GR4J) | Capacity of the nonlinear routing store (mm) | 326 |
| X4 (GR4J) | Unit hydrograph time base (day) | 3.50 |
| X5 (CemaNeige) | Cold content factor (-) | 0.004 |
| X6 (CemaNeige) | Snowmelt factor (mm/day/°C) | 3.66 |

- $\alpha$: ratio between the standard deviations of the simulated and observed streamflow time series; this quantifies the ability of the simulation to reproduce the variability of the considered variable, and ranges between 0 and $+\infty$ (values > 1 indicate a model overdispersion).

- $r$: coefficient of correlation between the simulated and the observed streamflow time series; this quantifies the ability of the simulation to reproduce the observed temporal variations of the considered variable, and ranges between -1 and 1 (perfect correlation).

Using KGE limits the biases of both water balance and variability, while keeping a good temporal correlation. Note that for each model simulation, the first simulated year was used as an initialization period, and was not considered for the final performance evaluation. All the rainfall-runoff model outputs presented in the manuscript have been produced at the daily resolution by using both GR4J rainfall-runoff model and its snowmelt routine CemaNeige.

### 3.3 Step 2: finding analogue dates (ANA)

The ANA approach is a resampling method based on synoptic circulation similarities between days, with a sampling of observed climatic series for a given time period (here, 1950-2011, *the observation period*) over a longer time period (here, the 1880-2011 period, *the reconstruction period*). The synoptic information considered for the analogy is geopotential height fields. Here, each day is described by four geopotential height fields: (i) 1000 hPa at 0h, (ii) 1000 hPa at 24h, (iii) 500 hPa at 0h, and (iv) 500 hPa at 24h. The geopotential height fields are extracted over a large domain covering the studied area (cf. sub-section 2.1.1, see Appendix A). The metric used to rank the days in terms of analogy is the Teweles and Wobus (1954) distance (see Appendix B), which highlights similarities in terms of geopotential field shapes (Obled et al., 2002), and has been shown to provide better outcomes than what is obtained by using classical Euclidean distances in this framework (Wetterhall et al., 2005). Note that a seasonal constraint is imposed for the identification of analogue days: the potential analogue days of a given day are the ones included in a 60-day period centered on the calendar studied day. Thus, analogues of a winter day are themselves winter days: for example, the potential analogue days for January 1, 1880 are all of the available days within

the December $1^{st}$ to January $30^{th}$ period of the observation period (here 1950-2011). Another constraint is also imposed for the identification of analogues in which no analogue can be selected if they are closer than 15 days from the chosen date. For example, the potential analogue days for January 1, 2000, are all of the available days within the December $1^{st}$ to January $30^{th}$ period of the observation period except the December 15, 1999 to January 15, 2000 period. The ranking of analogue days is
based on the Teweles-Wobus distance (see Appendix B). For each studied day, a given number of n analogues days is considered, thus generating a climatic ensemble of n time series. Here, the 20 nearest analogue days were selected for each studied day and each 20CR member considered ($n = 20$). Table 2 illustrates the generation of this climatic ensemble by giving several analogue days obtained for three particular dates (1880-01-01, 1880-01-02 and 2011-12-30). For example, when considering member 1 of the 20CR (M1), the first analogue day of 1880-01-01 is 1984-01-23, the second analogue day is 1991-12-12, and
the 20th analogue day is 1988-01-16. Finally, $20 \times 5$ (5 members of the 20CR considered) daily climatic series were generated over the 1880-2011 period.

### 3.4   Step 3: reconstruction of a daily climatic (P and T) ensemble

Using ANA outputs, ANATEM aims to exploit the available long-term reference time series (hereafter denoted as TEM) to improve the climatic reconstruction, by applying a classical regression between ANA outputs and the reference series. In
this study, the ANA approach was directly applied for the precipitation reconstruction (since no precipitation "witness" series was available), while the ANATEM approach was applied for the reconstruction of daily temperature (using the BEST daily temperature series). As in Kuentz et al. (2015), the local regression model (hereafter denoted as LM), applied here for the temperature reconstruction, is based on an additive correction, modeled by a daily harmonic function. The parameters of this regression function were estimated over the observation period (here, 1950-2011) on the interannual mean monthly residuals
of the differences between the catchment temperature series and the TEM series, and has the following expression:

$$\widehat{T}_{LM}(d) = T_{TEM}(d) + \beta(d) + \epsilon(d) \tag{2}$$

where $\widehat{T}_{LM}(d)$ is the estimate of the air temperature for the day $d$, $T_{TEM}(d)$ is the value of the witness series temperature for the same day, $\beta(d)$ is the correction, depending on the calendar day of the year, and $\epsilon(d)$ is a residual assumed to have zero mean.

The ANATEM method was applied at the daily resolution over the 1880-2011 period. The ensemble of temperature values reconstructed for the day $d$ has the following expression:

$$\widehat{T}_{ANATEM}^{k}(d)]_{k=1,...,n} = \widehat{T}_{LM}(d) + [T(d_k) - \widehat{T}_{LM}(d_k)]_{k=1,...,n} \tag{3}$$

where $[\widehat{T}_{ANATEM}^{k}(d)]_{k=1,...,n}$ is the ensemble of n reconstructed temperature values for the target day $d$, $\widehat{T}_{LM}(d)$ is the air temperature estimate obtained with the regression model for the day $d$, $d_k$ is the $k^{th}$ analogue day selected for the day $d$,
$T(d_k)$ is the observed temperature value for the $k^{th}$ analogue day, $\widehat{T}_{LM}(d_k)$ is the air temperature estimate obtained with the regression model for the $k^{th}$ analogue day, and $n$ is the total number of analogue days (here $n = 20$, see section 2.1.1).

**Table 2.** Illustration of the analogue dates obtained with the ANA approach. Here, a sub-sample of the 20 analogue days of three particular dates (1880-01-01, 1880-01-02 and 2011-12-30) are given for each of the five 20CR members considered (M1 to M5). The ranking of analogue days is performed with Teweles and Wobus (1954) distances.

| 20CR MEMBER | ANA | 1880-01-01 | 1880-01-02 | ... | 2011-12-30 |
|---|---|---|---|---|---|
| M1 | ANA1 | 1984-01-23 | 1959-02-13 | ... | 2007-12-18 |
| M1 | ANA2 | 1991-12-12 | 1961-01-11 | ... | 1989-11-05 |
| M1 | ... | ... | ... | ... | ... |
| M1 | ANA20 | 1988-01-16 | 1953-12-25 | ... | 2007-12-19 |
| M2 | ANA1 | 1984-01-23 | 1974-12-27 | ... | 1979-11-19 |
| M2 | ANA2 | 1990-11-30 | 1961-01-11 | ... | 1971-11-13 |
| M2 | ... | ... | ... | ... | ... |
| M2 | ANA20 | 1957-02-02 | 1990-02-19 | ... | 1976-12-04 |
| M3 | ANA1 | 1950-02-03 | 1950-02-04 | ... | 2007-12-18 |
| M3 | ANA2 | 1989-01-13 | 1971-12-24 | ... | 1989-11-05 |
| M3 | ... | ... | ... | ... | ... |
| M3 | ANA20 | 1990-11-30 | 1957-02-07 | ... | 2003-12-14 |
| M4 | ANA1 | 1986-12-15 | 1956-12-21 | ... | 2007-12-18 |
| M4 | ANA2 | 2007-01-02 | 1974-01-19 | ... | 1989-11-05 |
| M4 | ... | ... | ... | ... | ... |
| M4 | ANA20 | 2004-12-29 | 1971-12-24 | ... | 1994-11-20 |
| M5 | ANA1 | 1984-01-23 | 1961-01-11 | ... | 2007-12-18 |
| M5 | ANA2 | 1989-01-13 | 1962-01-25 | ... | 1971-11-13 |
| M5 | ... | ... | ... | ... | ... |
| M5 | ANA20 | 1993-11-09 | 1965-11-04 | ... | 1958-11-16 |

The final climatic ensemble is built with 100 precipitation (ANA outputs) and air temperature (ANATEM outputs) daily series over the 1880-2011 period. For each day, the 100 climatic values are obtained based on the 20 "closest" analogue days for each of the 5 20CR members considered.

### 3.5   Step 4: reconstruction of a daily streamflow ensemble

Using the rainfall-runoff model parameter set obtained after calibration (step 1), the reconstructed climatic ensemble is finally transformed into one streamflow ensemble, available over the 1881-2011 period (1880 being used as an initialization period) at the daily temporal resolution. The final streamflow ensemble thus consists of 100 daily streamflow series over the 1881-2011 period.

### 3.6 Comparison of reconstructed series against observations

In order to compare the reconstructed streamflow time series against observations, the reconstructed ensembles were first aggregated: a daily series was generated for each of the five 20CR members considered by averaging the 20 daily series constituting each ensemble. The five daily mean series are denoted as $\overline{ANA}$ or $\overline{ANATEM}$, depending on the method used to produce them. The evaluation of the reconstruction performances was based on the three KGE components and its final values. For the reconstructed climatic time series, the computation of these four scores was carried out over the 1950-2011 period, at the daily time scale but also at the monthly time scale, in order to evaluate the intra-annual reconstruction performances, and at the yearly time scale, in order to evaluate interannual reconstruction performances. For the reconstructed streamflow ensemble, these scores were computed over mean annual flow values and mean May flow values over two time periods, 1963-1979 (rainfall-runoff model calibration period) and 1982 to 2011 (naturalized flows).

## 4 Results

### 4.1 Rainfall-runoff model calibration performances (1963-1979

Over the 1963-1979 calibration period, the CemaNeigeGR4J model performs really well with a KGE value of 0.93 (rainfall-runoff simulation with KGE > 0.8 are generally considered as good). The values of the 6 calibrated parameters are detailed in the Table 1. Figure 4 presents the performance of the CemaNeigeGR4J rainfall-runoff model over the calibration period (1963-1979). Simulated and observed quantiles of monthly streamflow show a strong correlation (Fig. 4a), with a limited over-estimation of the lowest values by the rainfall-runoff model observed during the winter months (from January to April, Fig. 6b). The timing of the simulated regime is similar to the observed one. However, systematic limited biases are found, with an over-estimation of the winter streamflow values (January to April) and of the spring flood values (June) and an underestimation of the streamflow values during the snowmelt period (July to October).The model is also able to simulate the general interannual variability of mean annual streamflow (Fig. 4c), with higher values for the 1964-1969 period and lower values for the 1970-1976 period, for example. Nevertheless, non-systematic biases are found for several years, with both underestimations (e.g., 1964 and 1969 years) and overestimations (e.g., 1972 and 1975 years) of mean annual streamflow values. Finally, the observed and modeled distributions of annual streamflow values are similar (Fig. 4d), with an overestimation of the lowest mean annual streamflow values.

### 4.2 Climatic reconstructions (1950-2011 and 1880-2011

In this section, the results of the climatic reconstruction are presented, first in terms of performance estimated over the observed period (1950-2011), and then in terms of centennial mean annual series (1880-2011).

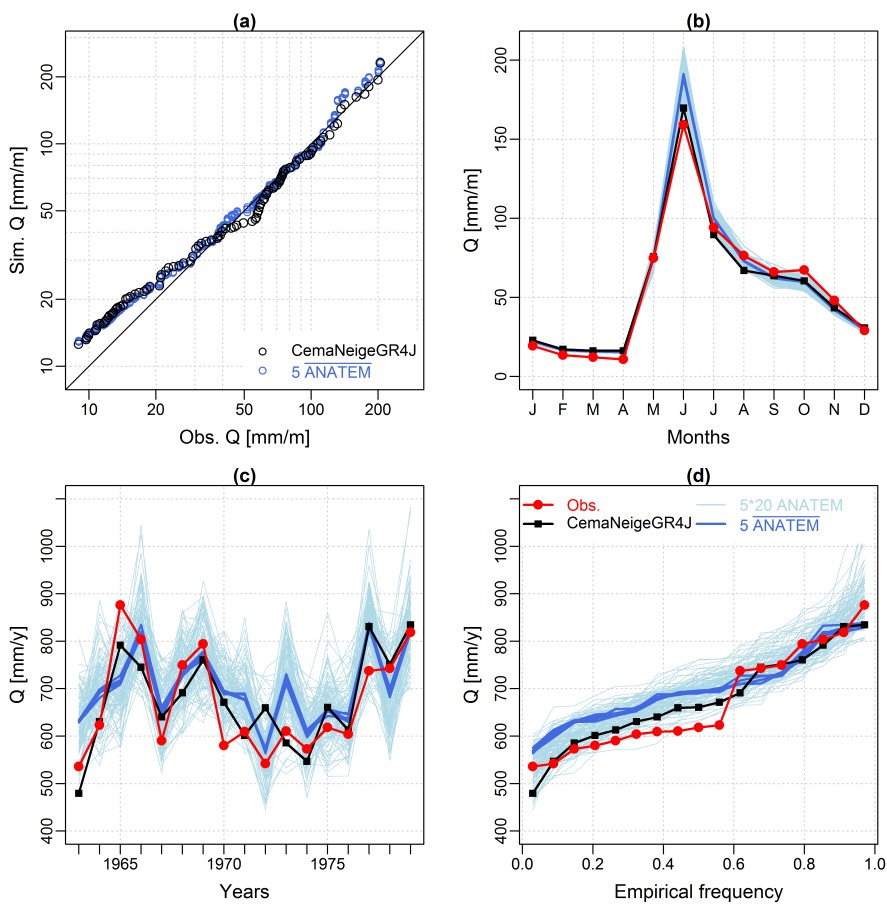

**Figure 4.** Performances of the CemaNeigeGR4J rainfall-runoff model (black) and of the ANATEM flow reconstruction (blue colors) evaluated over the calibration period of the rainfall-runoff model (1963-1979). (a) Monthly quantile-quantile plots (logarithmic scale), (b) observed and simulated monthly streamflow regime, (c) observed and simulated interannual streamflow variability, and (d) observed and simulated streamflow yearly mean distribution. The legend indicated on the (d) graph is also valid for the (b) and (c) graphs

### 4.2.1 Performance of the climatic reconstructions over the observation period (1950-2011)

Figure 5 compares the temperature reconstruction (using ANA and ANATEM outputs) and precipitation reconstruction (using ANA outputs) to the observations for the 1950-2011 period, in terms of monthly regimes and yearly value distributions. For temperature, the ANATEM reconstruction is excellent, both in terms of monthly regime and yearly mean value distribution. The ANA temperature reconstructions (in grey) show a limited performance for the coldest months (December and January) and for the warmest months (July and August), and thus highlight the importance of using the BEST temperature series through

ANATEM, which successfully corrects the ANA outputs. The intra-variability of the ANATEM temperature ensemble is very limited.

The precipitation reconstruction is not as good as that of the temperatures. The timing of the monthly regime is well captured, with lowest monthly precipitations observed in February, and the highest in July. However, an overestimation of the reconstructed precipitation is observed for all months, with the exception of January and September. Overall, a wet monthly bias of precipitation is found. This bias is also seen in the plot of the yearly value distributions (Fig. 5d), which show that a majority of the mean annual precipitation values are overestimated by the reconstruction. In terms of variability within the ensemble, the similarity of the five 20CR members $\overline{ANA}$, in blue) shows that the uncertainty of the geopotential height field (quantified here through the consideration of the five members) has a negligible impact on the precipitation reconstruction over this time period and at these resolutions (yearly and monthly). The relatively large width of the ANA ensembles (grey envelopes) indicates that the uncertainty due to the selection of 20 analogue days has an impact on the precipitation reconstruction.

Figure 6 summarizes the climatic reconstruction performances at the daily, monthly and yearly resolutions, both over the 1950-2011 period. For air temperature (Fig. 6a), and as previously indicated, the overall reconstruction performances are excellent for ANATEM outputs (KGE > 0.9), and limited for ANA outputs (KGE > 0.4). ANA outputs (grey points) are characterized by an overestimation ($\beta > 1$) tendency for the three resolutions and an underdispersion ($\alpha < 1$) tendency for the monthly and yearly resolutions. If the yearly temporal correlation is good at the daily and at the yearly resolutions, the temporal correlation is excellent at the monthly resolution (r $\approx$ 1). For precipitation (Fig. 6b), the overall reconstruction performance is better at the monthly resolution (KGE > 0.6) than at the daily (KGE ranging between 0.3 and 0.5) and yearly resolutions (KGE ranging between 0.2 and 0.6). The reconstructed time series show a clear overestimation bias, an underdispersion problem, and a limited temporal correlation at the three different resolutions. Averaging each ensemble of the considered 20CR members (blue points) results in better temporal correlations at the daily and yearly resolutions, but at the expense of a too small reconstructed variability.

### 4.2.2 Centennial mean annual climatic series (1880-2011)

Figure 7 shows the reconstructed climatic series over the entire studied period (1880-2011), at the yearly resolution. For temperature, the ANATEM reconstruction shows a very good fit to the observed series, with the exception of the first decade (1950-1960), when the reconstructed annual temperatures appear to be systematically lower than the observed annual temperature. ANA ensembles are larger than their ANATEM counterparts, and perform worse in terms of mean annual temperature variability. The good performance of the ANATEM reconstruction is largely due to the BEST series, which is strongly correlated with the observed series at the annual resolution, except for the first observed decade. At the centennial scale, the reconstructed temperature time series are highly similar to the BEST series, showing that the entire temperature signal reconstructed is driven here by the BEST series. The ANATEM ensemble width is narrow at the annual time scale, as has already been seen for the monthly regime (Fig. 5a and b). The reconstruction shows an increase in the Caniapiscau catchment mean annual temperature over the last 130 years.

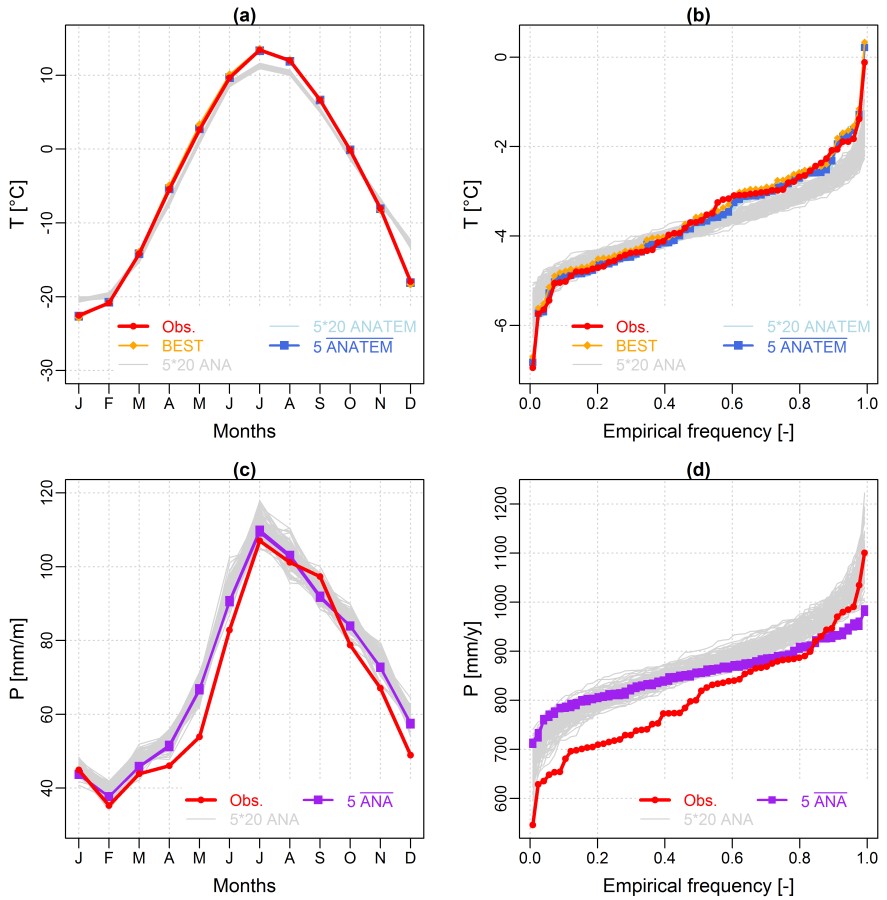

**Figure 5.** Monthly regimes (a and c) and yearly value distributions (b and d) for temperature (with ANA and ANATEM) and precipitation (with ANA) reconstructions and observations over the 1950-2010 period. Note that for temperature monthly regime (a), the ANATEM simulations are similar to the observations, and thus, ANATEM curves (blue) are not visible since they are below the observation curve (red).

For mean annual precipitation, the ANA reconstruction does not perform as well, especially over the last two decades (1990-2010), where the reconstruction failed to reproduce the observed low values for the mean annual precipitation (compared to mean values over the entire observed period). A similar bias is found for the 1950-1965 period, while the variability of the mean annual precipitation values during the 1965-1985 period are well reproduced. Relatively, the precipitation reconstruction seems to be able to reproduce the wet-dry periods, but fails to match the observed values. Considering the reconstruction at the centennial time scale, no significant trend is found for mean annual precipitation. Several periods are interesting, such as the sequence of wet and dry years around 1920. Finally, variability due to consideration of five 20CR members is seen until 1940, and seems to be higher for several time periods, such as the 1880-1890 decade.

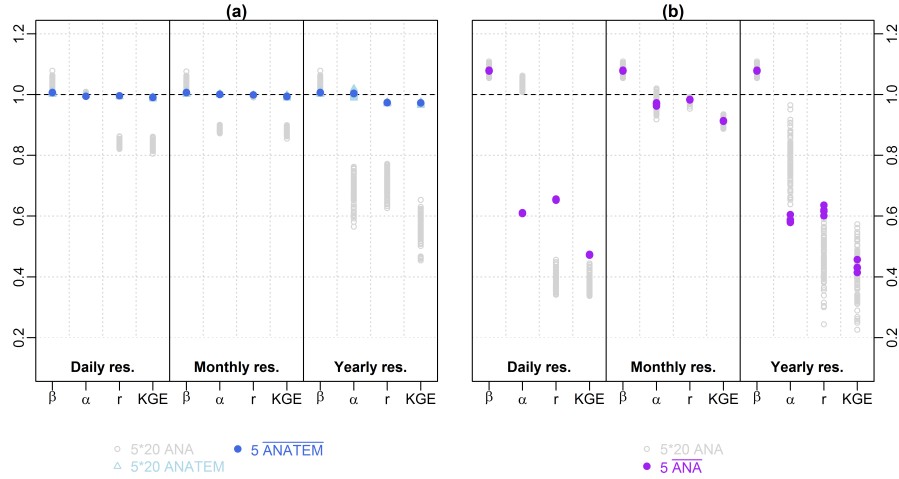

**Figure 6.** Daily, monthly and yearly performances of the air temperature ANA and ANATEM reconstructions (a) and the ANA precipitation reconstructions (b), for 1950-2011 period.

## 4.3 Streamflow reconstructions (1962-2011 and 1881-2011)

In this section, the results of the streamflow reconstructions are presented, first in terms of performance estimated over two time periods, and then in terms of centennial series (annual mean flows and spring flood values).

### 4.3.1 Performance of streamflow reconstruction over two observed periods (1962-2011

Using the five climatic ensembles produced by ANA (for precipitation) and ANATEM (for temperature) as inputs to the CemaNeigeGR4J rainfall-runoff model, five ensembles of 20 daily streamflow series were produced over the 1881-2011 period (the year 1880 is used as an initialization period for the rainfall-runoff model). Figure 4 presents the performance of the streamflow reconstructions over the rainfall-runoff model calibration period (1963-1979). The obtained reconstructions have, logically, the same qualities and defaults characterizing the climatic reconstructions (presented in section 4.2.1) and the rainfall-runoff model performance (presented in section 4.1). Figure 4a is a quantile-quantile plot between observed and simulated mean monthly streamflows. Monthly correlations between observations and simulations are good, but reveal a systematic overestimation of the lowest mean monthly streamflow values (winter months). A clear overestimation of the monthly flood peak (June) is also found (cf. Fig. 4b), due both to the rainfall-runoff model performance on this catchment and a general overestimation of the precipitation by the climatic reconstruction, as already shown in Fig. 5. Observed and simulated interannual variabilities are similar, but with an overestimation of the mean annual streamflow values by the reconstructions, especially for the years with relatively low mean annual streamflow values (1971-1976).

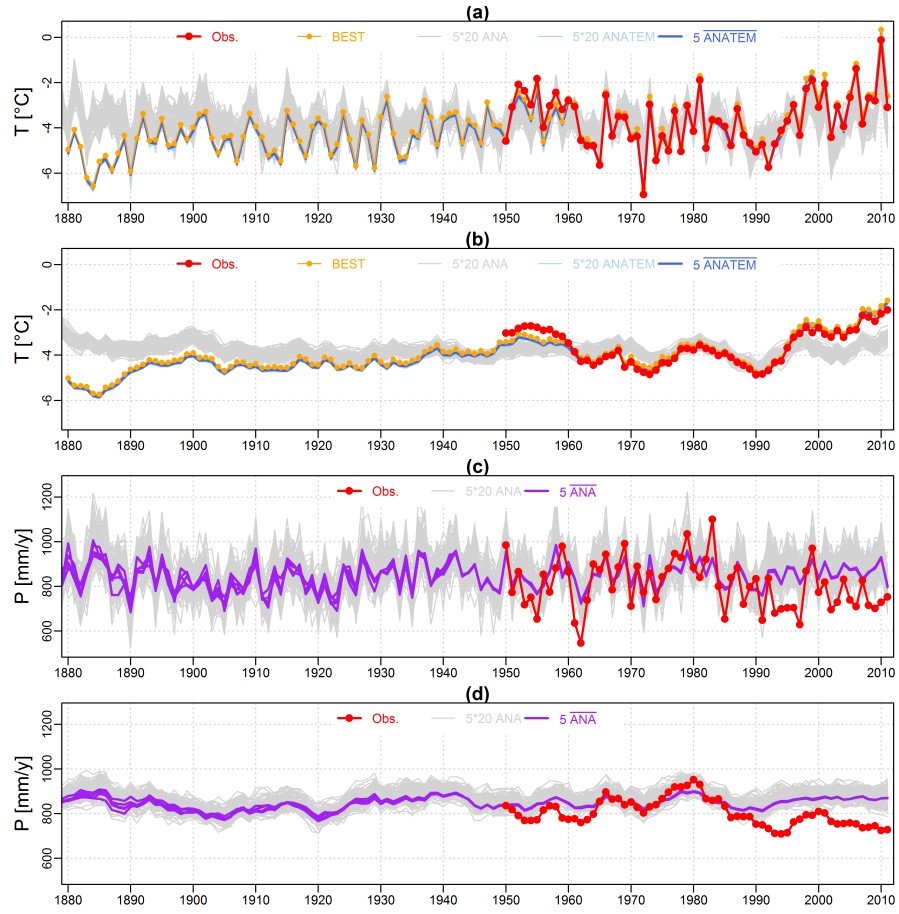

**Figure 7.** Interannual variability of reconstructed mean annual values of temperature (ANA and ANATEM outputs) and precipitation (ANA outputs) compared with observations over the 1880-2011 period. (a) and (c) are raw yearly values while (b) and (d) are 6-year running means of mean annual temperature and mean annual precipitation, respectively.

Figure 8 summarizes the performances of the streamflow reconstructions over two periods (1962-1979 and 1981-2011), in terms of mean annual streamflow values (Fig. 8a) and May monthly flow values (Fig. 8b). Overall KGE performances are limited to good for mean annual streamflow series and very good for the May monthly flow series. Again, an overestimation of mean annual flows is found for both periods. For May monthly flows, no specific trend is found for the first period, while a slight underestimation is observed for the second period. The performances of the dendrohydrological reconstructions are also evaluated and are shown in Figure 8, highlighting that dendrohydrological reconstructions perform slightly better than ANATEM ones for the mean annual streamflow values while ANATEM reconstructions perform better than dendrohydrological ones for the May monthly flow values.

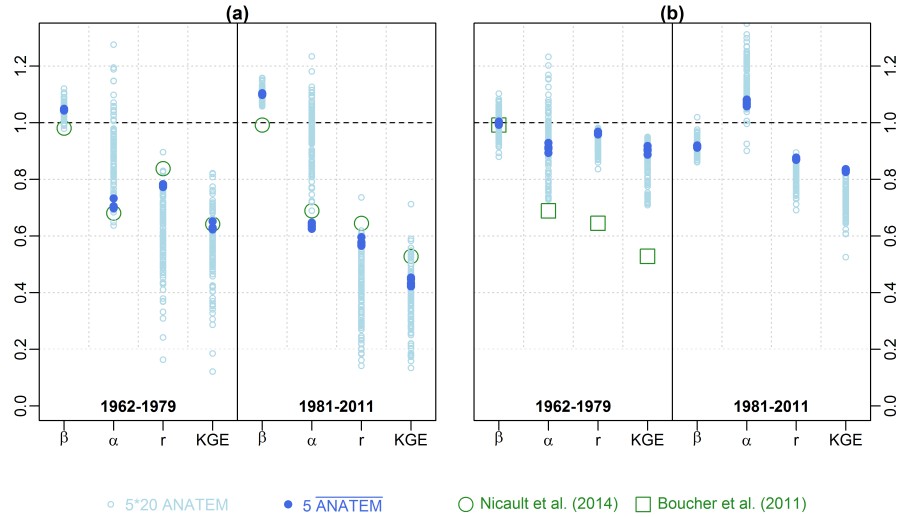

**Figure 8.** Streamflow reconstruction performances evaluated over two periods (1962-1979 and 1981-2011), (a) mean annual streamflow values and (b) May monthly flow values. Dendrohydrological reconstruction performances are also evaluated over the 1962-1979 and 1982-2001 periods for mean annual streamflow values (a) and the 1962-1979 period for May monthly flow values (b).

### 4.3.2 Centennial mean annual flow reconstructions (1881-2011)

Figure 9 presents the centennial ANATEM streamflow reconstruction and compares the reconstruction to observations and to the mean flow reconstruction proposed by Nicault et al. (2014) using tree rings. As shown in Fig. 4, a good correlation is found between the ANATEM reconstruction and observations for the 1963-1979 period. Considering the other streamflow
5   observation time period (naturalized flows of 1982-2011), the correlation is weaker, with a general overestimation of the mean annual streamflow. At the centennial scale, a comparison between ANATEM and tree ring mean flow series reveals that the two series are not statistically different, since the ANATEM ensemble is within the tree ring confidence interval (green envelopes), except for the 1930-1940 period. For this period, and especially around 1940, ANATEM mean flow reconstructed values are significantly higher than tree ring ones. A significant variability of mean annual streamflow is simulated for the 130 past
10  years. The two reconstructions agree for the 1880-1910 period, simulated as a period of decreasing mean annual streamflows, followed by a 10-year increasing period. The 1920-1950 period shows differences between the two reconstructions, with ANATEM mean flows being larger than for tree rings. For the 1950-2011 period, the mean flow relative evolutions are similar, but the absolute values are different, with ANATEM values being systematically higher than tree ring values. This constant bias could be explained by the overestimation of precipitation over the record period. The 1912 year seems to be a "hydrologically
15  interesting year", since it is simulated as a very wet year by tree rings, while simulated as a dry year by ANATEM. Finally,

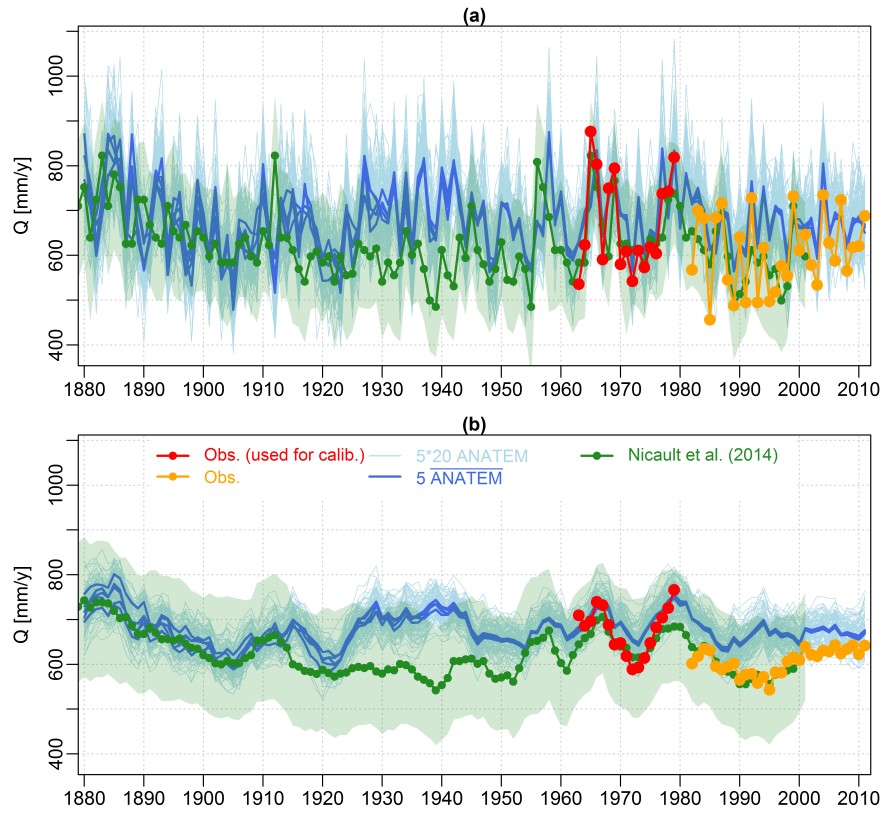

**Figure 9.** ANATEM mean flow reconstructions: comparison with observations and Nicault et al. (2014) tree ring series, 1881-2011 period. (a) is raw yearly values while (b) is 6-year running means of mean flows.

as for the ANA precipitation reconstruction, the variability due to consideration of five 20CR members is seen until the year 1940, and seems to be higher over the distant past.

### 4.3.3 Centennial spring flood reconstruction (1881-2011)

Finally, Fig. 10 presents the ANATEM centennial spring flood reconstruction compared to observations and to the reconstruction proposed by Boucher et al. (2011) using tree rings. For ANATEM and for the observed streamflow series, these annual series were constituted by estimating, for each year, the May monthly flow, since Boucher et al. (2011) produced a May streamflow reconstruction. The correlations between the ANATEM reconstruction and the observed series (1963-1979 and 1982-2011) are excellent and very good, respectively, and thus reproduce the increase of spring floods during the 1970-1980 period, and then the decrease during the 1980-1990 period, finally followed by a slight increase and a stagnation over the two last decades. At the centennial scale, the two reconstructions appear to be significantly different for a long period of time, since the ANATEM ensemble is out of the tree ring confidence interval for the 1881-1920 period. Another significant difference

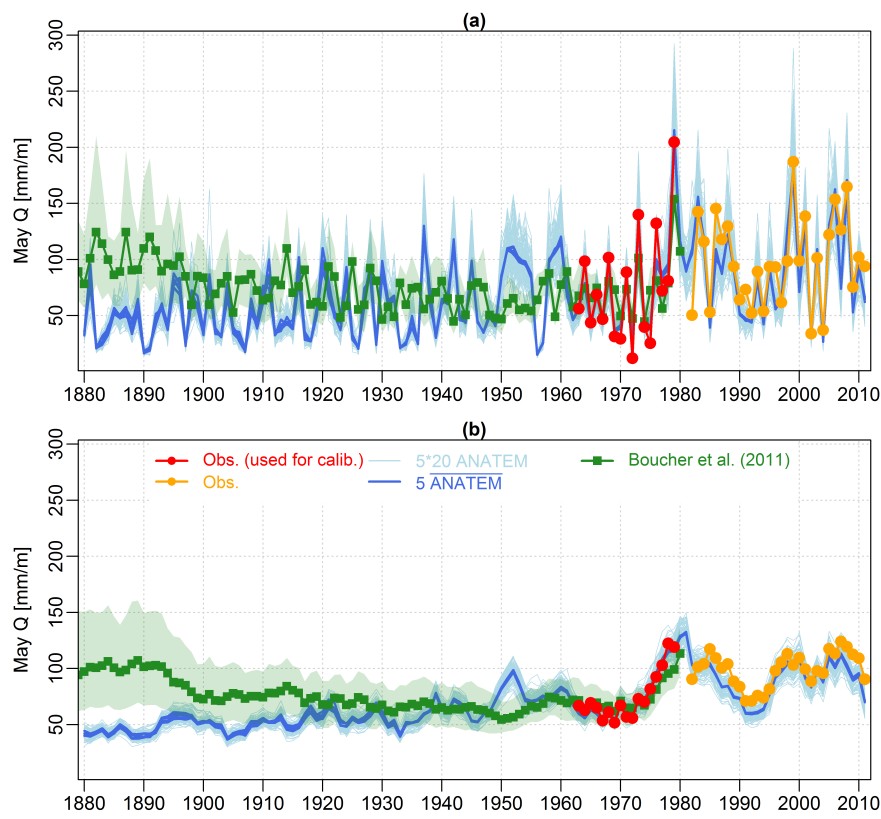

**Figure 10.** ANATEM spring flood reconstructions: comparison with observations and Boucher et al. (2011) tree ring series, 1881-2011 period. (a) is raw yearly values while (b) is 6-year running means of spring flood values.

exists over the 1950-1960 period, seen as a common decade by the tree ring reconstruction (reconstructed spring flood ranging from 47 to 87 [mm/m]), while being seen as a highly variable hydrological decade for the ANATEM reconstruction, with high values for the first five years (around 110 [mm/m] for the 1950-1955 period), and then two very low values (around 20 [mm/m] for the 1956-1957 period), finally followed by three high value years (around 110 [mm/m] for the 1958-1960 period). Overall, the ANATEM reconstruction simulated an increasing trend of spring floods for the Caniapiscau catchment. This trend is related to the increasing temperature trend, as illustrated in Fig. 7.

## 5 Discussion and conclusion

In this study, a daily hydro-climatic reconstruction is proposed for the Caniapiscau Reservoir (northern Quebec, Canada) for the 1881-2011 period. This reconstruction was generated by firstly applying the ANATEM method (Kuentz et al., 2015), combining large-scale atmospheric information (here the NOAA 20th Century geopotential height reanalysis, (Compo et al.,

2011)) with local climatic observations - when such series are available - to produce a daily ensemble of climatic series (precipitation and air temperature). Secondly, this climatic ensemble was used as input to a rainfall-runoff model (here GR4J (Perrin et al., 2003) and its snow accumulation and melt routine CemaNeige (Valéry et al., 2014a) previously calibrated in order to obtain a streamflow ensemble, at the daily resolution. The performances of the climatic reconstructions were quantified over the observed period (1950-2011) and showed very good performance for air temperature, both in terms of monthly regime and interannual variability. This excellent performance is due mainly to the use of a local reference temperature time series (here, a daily temperature time series extracted from the Berkeley Earth Surface Temperature analysis, Rohde et al. (2013)). For precipitation, no local reference climatic time series was available and the precipitation reconstructions are thus only a function of geopotential height field analogy. The precipitation reconstructions present a good performance in terms of regime, but with a somewhat limited ability to reproduce the observed annual values and interannual variability, combined with a systematic wet bias. The performance of the streamflow reconstruction was then compared to streamflow observations. This comparison showed a good performance, both in terms of monthly regimes and interannual variability, with a systematic overestimation of the mean annual streamflow values, due mainly to the wet bias of the precipitation reconstruction by the ANATEM method.

These newly produced reconstructions were then compared to two different reconstructions performed on the same catchment by using tree ring data series, one being focused on mean annual flows (Nicault et al., 2014), and the other on spring floods (Boucher et al., 2011). In terms of mean annual flows, the interannual variability of flows reconstructed by tree rings and ANATEM were similar (except for the 1930-1940 decade), with significant changes seen in wetter and drier years. This variability seemed to be driven mainly by the variability of mean annual precipitation. In terms of spring floods, the interannual variabilities reconstructed by tree rings and by ANATEM were quite similar for the 1955-2011 period, but significantly different for the 1880-1940 period. The ANATEM spring flood reconstruction showed an increasing trend over time, and this variability seemed to be driven by the variability of the mean annual temperature.

These results emphasize the need to apply different reconstruction methods on the same catchments. Indeed, such comparisons highlight potential differences between available reconstructions, and finally, allow a retrospective analysis of the proposed reconstructions of past hydro-climatological variabilities. In this study, two very different reconstruction methods were applied on the same catchment, revealing several periods where the two reconstructed streamflow series differ considerably. Thus, in terms of mean annual flows, the year 1922 and the 1930-1940 decade appear to be particularly dry and wet, respectively, when reconstructed with the ANATEM method, while they are simulated as particularly wet and dry when reconstructed using tree ring proxies. In terms of spring floods, the two reconstruction methods are in disagreement for the 1950-1960 decade, simulated as a decade with wide variabilities by ANATEM, with short sequences of alternating high and low spring flood values, compared to the tree ring reconstruction. Further investigation is needed in order to understand the differences for these specific periods. Finding indications of particular hydro-climatic conditions at the regional scale through the analysis of documents, reports or ad-hoc measurements could represent a means of assessing the respective performances of each reconstruction method. More generally, the long-term signals of the spring flood reconstructions are different, with a clear increasing tendency for floods reconstructed with ANATEM, related to the mean annual temperature rise in this region through the studied decades. Further work is needed to investigate this difference between the two reconstructions.

The evaluation of the analogue performance revealed two main limitations for the precipitation reconstruction. Firstly, a general wet bias was found when the reconstructed precipitation time series were compared to observations, and therefore, a similar bias was observed for streamflow reconstruction. A classical bias-correction method could be applied on the reconstructed precipitation time series in order to eliminate this bias. However, applying a bias correction method implies an additional error source which could be amplified when the streamflow is analyzed (Teng et al., 2015), and, even more importantly, raises the issue of the bias stationarity (e.g., Teutschbein and Seibert, 2013; Chen et al., 2015; Velázquez et al., 2015). Secondly, the interannual variability of mean annual precipitation is reproduced with limited performances on the Caniapiscau reservoir catchment. The inability of the analogue approach to reproduce the interannual precipitation variability - already highlighted by Kuentz et al. (2015) over 22 French catchments – is due to the absence of a local reference climatic time series, unlike for temperature reconstruction, where a local temperature time series is used, and ensures that the simulated interannual temperature variability is reproduced efficiently. Finding an additional series which significantly improves the precipitation reconstruction is a major perspective of this work. The use of variables produced by the available reanalyses (e.g., relative humidity, precipitable water content) for finding analogue dates will be investigated, along with the testing of time series of local pressure measurements. For example, Caillouet et al. (2016) showed that adding the sea surface temperature variable to the temperature, geopotential, vertical velocity and humidity for finding analogue dates significantly improves the reconstruction of air temperature and precipitation over France.

In this study, most of the ANA approach options used to find analogue days were defined by looking at previous applications of the same methodology (e.g., Horton et al., 2012; Chardon et al., 2014) and by sensitivity analyses (results partially shown in Appendix A). The sensitivity of the final reconstructions to these options (size of the geopotential height domain extension (see Appendix A), choice of the geopotential height levels studied, number of analogue days, etc.) could be further investigated in a future work. Interestingly, the uncertainty due to the use of five members of the 20CR reanalysis appears to be limited, and even null from 1940 onward. See for example Fig. 9 which presents the centennial ANATEM streamflow reconstructions: it is impossible to distinguish the five ANATEM average series after 1940, highlighting that considering five different members of the 20CR reanalysis as inputs of the reconstruction method has a negligible impact on the reconstruction of the mean annual streamflow.

Finally, the reconstructed climatic time series are transformed into streamflow time series thanks to a daily rainfall-runoff model, previously calibrated over the relatively short observation period (whith really good calibration performances). The use of one model, one objective function and one parameter set is questionable. Quantifying the sensitivity of the obtained reconstruction to the hydrological modeling assumptions made was out of the scope of this paper, but definitively deserves further research, especially considering the issue of uncertainty due to rainfall-runoff model parameters in a changing climate. Thus, numerous authors highlighted that calibrated parameters of rainfall-runoff models are dependent on the climate of the calibration period and that performance decreases when applied over periods where the climate differs from that of calibration period (e.g., Merz et al., 2011; Coron et al., 2012; Brigode et al., 2013b). Thus, testing different calibration strategies (e.g., bootstrap calibration used by Brigode et al. (2015)), testing particular objective functions especially devoted to the final study objective (e.g., studying mean annual streamflow), and adapting the time step of the rainfall-runoff model to the objective

would be interesting for future works.

The combination of the ANATEM reconstruction method with a rainfall-runoff model offers an interesting method for use in reconstructing hydro-climatic time series at a very fine resolution (here daily), which is usually needed in applying impact
models (such as dam management models), and finally, to discuss the climatic process, which significantly influences the hydrological decadal variability at the catchment scale. An interesting perspective would be to test this modeling approach on numerous other catchments, and focusing on regions where long and good quality hydro-climatic time series are available, thus giving the opportunity to quantitatively evaluate the reconstruction methodology over long time periods. Kuentz et al. (2013) thus reconstructed 110-year streamflow time series for 22 French catchments with a combination of the ANATEM reconstruc-
tion method and a daily rainfall-runoff model, reconstitutions which allowed to discuss the hydro-climatic variability over the last century in the studied region (French Alps). Finally, these applications could also give interesting insights on regions where it is not sufficient to consider only climatic time series in explaining observed multi-decadal hydrological variability, and thus highlight other significant factors influencing hydrological variability that need to be quantified (e.g., changes in land use, urbanization or hydrogeology).

Another way to evaluate the two reconstruction methods would be to use the hydro-climatic time series reconstructed by ANATEM as inputs for a tree diameter growth model (e.g., models developed and applied for black spruces (*Picea mariana* [Mill.] BSP) in Canada by Subedi and Sharma (2013) and Huang et al. (2013)), and to then compare the tree ring simulated through this growth model with the observed tree ring series.

**Appendix A**

Several tests have been performed for choosing the spatial domain to consider for the description of the geopotential height fields (see Brigode et al. (2013a); Radanovics et al. (2013) for similar approaches). Here, eight different spatial domains have been tested (domain numbered from 1 to 8). These domains, illustrated on Fig. 11a, are centered on the Caniapiscau catchment and are of progressively larger. For each domain, a climatic reconstruction has been performed with the ANA method for the Caniapiscau catchment but also for 211 other Quebec catchments of the cQ2 database Guay et al. (2015). These recon-
structions have been performed on the 1990-2010 period with only one member of the 20CR reanalysis. The performances of these different reconstructions have been evaluated by comparing observed series with reconstructed series looking at different precipitation and air temperature criterion. Fig. 11b presents the three criterion chosen to evaluate the precipitation reconstruction: (i) the correlation between observed and reconstructed annual precipitation series (first line, optimal value is 1), (ii) the correlation between observed and reconstructed daily precipitation series (second line, optimal value is 1) and (iii) the
bias between observed and reconstructed precipitation series (last line, optimal value is 0). The boxplots summarize the performances obtained over the 211 catchments, while the purple point highlights the performance obtained specifically over the Caniapiscau catchment. Domain n°5 was finally chosen as a (subjective) compromise between having high correlation between reconstructed and observed precipitation series (at yearly and daily resolutions) and having low precipitation bias between re-

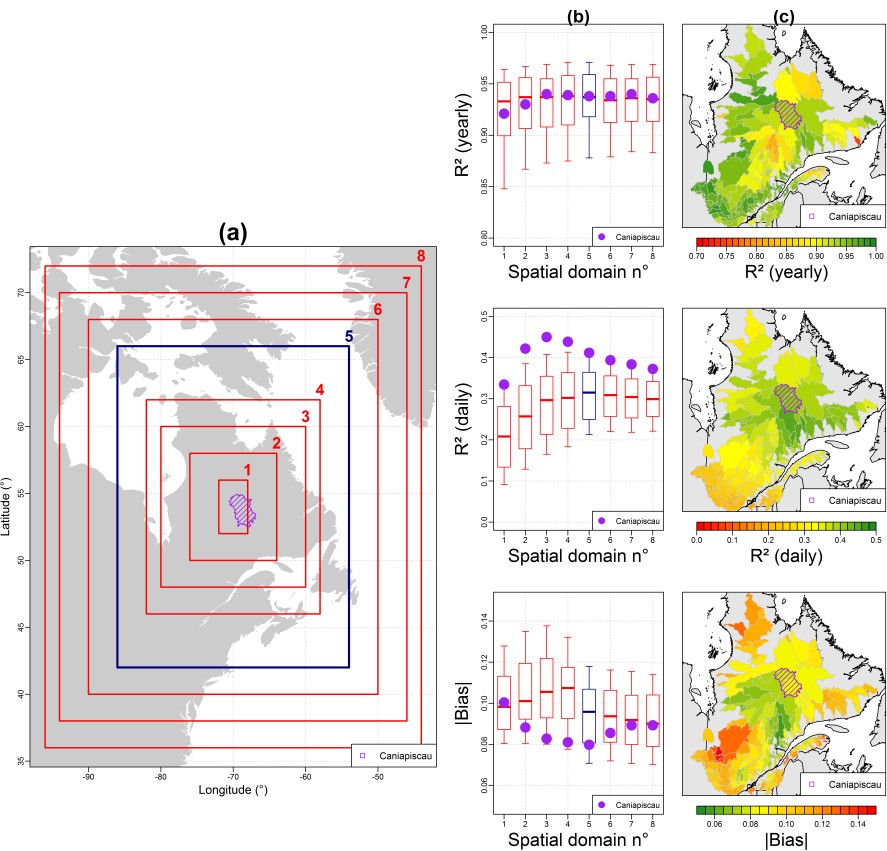

**Figure 11.** (a) Spatial extension of the eight geopotential height domains considered. (b) Performances of the precipitation ANA reconstruction estimated over the 1990-2010 period for 211 catchments of the cQ2 database (the boxplots are constructed with the 0.10, 0.25, 0.50, 0.75 and 0.90 percentiles). (c) Spatial distribution of the performances obtained with the domain n°5 over the 211 catchments of the cQ2 database. The Caniapiscau catchment is highlighted with purple color.

constructed and observed series on both the studied catchment (Caniapiscau) and on other neighboring Quebec catchments. Thus, we believe that the methodology performed in this study could also be used for the reconstruction of streamflow series on other neighboring catchments. Finally, Fig. 11c presents the spatial distribution of the three criterion values obtained within domain n°5. These maps reveal interesting spatial patterns, highlighting for example higher performances in terms of daily
5    precipitation correlation obtained for northern catchments compared to southern catchments. It is out of the scope of this paper to discuss the spatial variability and the spatial patterns of the climatic reconstruction performances, but this issue definitively deserves further research.

## Appendix B

The Teweles and Wobus (1954) distance (noted $D_{TW}$ hereafter) is used to find analogues to the synoptic circulation of a given day and thus to quantify the (di)similarity between two synoptic spatial configurations, each characterized by by four geopotential height fields over a given spatial domain (see Appendix A): (i) 1000 hPa at 0h, (ii) 1000 hPa at 24h, (iii) 500 hPa at 0h, and (iv) 500 hPa at 24h. The final $D_{TW}$ between a day A and another day B is the sum of four $D_{TW}$ calculated for each of the four geopotential height fields. The distance between the geopotential height field Z (e.g. 1000 hPa at 0h) of the day A and the day B is calculated as follow:

$$D_{TW,Z} = 100 \times \frac{\sum_{i=1}^{I-1}\sum_{j=1}^{J}\left|\Delta_{i,j}^{i,A} - \Delta_{i,j}^{i,B}\right| + \sum_{i=1}^{I}\sum_{j=1}^{J-1}\left|\Delta_{i,j}^{j,A} - \Delta_{i,j}^{j,B}\right|}{\sum_{i=1}^{I-1}\sum_{j=1}^{J}\max\left(\left|\Delta_{i,j}^{i,A}\right|,\left|\Delta_{i,j}^{i,B}\right|\right) + \sum_{i=1}^{I}\sum_{j=1}^{J-1}\max\left(\left|\Delta_{i,j}^{j,A}\right|,\left|\Delta_{i,j}^{j,B}\right|\right)} \tag{B1}$$

With:

- $\Delta_{i,j}^{i,A} = Z_{i+1,j}^{A} - Z_{i,j}^{A}$ is the geopotential gradient of a west-east direction starting from a point (i,j) for the day A.

- $\Delta_{i,j}^{j,A} = Z_{i,j+1}^{A} - Z_{i,j}^{A}$ is the geopotential gradient of a south-north direction starting from a point (i,j) for the day A.

This distance is thus focused on the synoptic circulation gradients (south-north and west-east directions) and not on the absolute geopotential height values. $D_{TW}$ ranges from 0 (for two identical fields) and 200 (for two opposite fields).

*Acknowledgements.* Support for the Twentieth Century Reanalysis Project, version 2c dataset, was provided by the U.S. Department of Energy, Office of Science Biological and Environmental Research (BER), and by the National Oceanic and Atmospheric Administration Climate Program Office. The authors thank the two reviewers and the editor who provided constructive comments on an earlier version of the manuscript, which helped clarify the text.

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
