# Peer review of "Streamflow variability over 1881-2011 period in northern Quebec: comparison of hydrological reconstructions based on tree rings and on geopotential height field reanalysis"

_Climate of the Past, 2016_

## Referee Comment (RC1) · Anonymous Referee #1 · 16 Feb 2016

Brigode et al.: Streamflow variability over 1881-2011 period in northern Quebec: comparison of hydrological reconstructions based on tree rings and on geopotential height field reanalysis

This manuscript provides a description and evaluation of streamflow reconstruction for the Caniapiscau Reservoir in northern Quebec (Canada) using a hydroclimatic reconstruction method that involves geopotential height reanalysis data, air temperature reanalysis data from the Berkeley Earth Surface temperature (BEST) analysis, an analogue approach to develop air temperature and precipitation data for the region, and

rainfall-runoff modeling with the GR4J model. The interesting approach is of value for presentation and discussion. In regards to the items requested by the journal:

1. Does the paper address relevant scientific questions within the scope of CP? The paper is about a reconstruction of streamflows, so it seems appropriate.

2. Does the paper present novel concepts, ideas, tools, or data? The approach presented does seem novel and, if edited to be more clear and complete, should be a useful presentation of the tools used.

3. Are substantial conclusions reached? The authors reach conclusions that are not always clear but are probably useful and substantial. See additional comments on manuscript.

4. Are the scientific methods and assumptions valid and clearly outlined? Some of the methods are adequately presented, but as someone unfamiliar with the techniques the authors used, I found it difficult to follow the methods at times. In addition, very little information is provided on the rainfall-runoff model so that the use of that model is not adequately justified and validated in the manuscript.

5. Are the results sufficient to support the interpretations and conclusions? The results presented seem necessary to support most of the interpretations and conclusions. As stated above, more information is needed regarding the rainfall-runoff model to support the interpretations and conclusions. I also found it difficult to follow a lot of the arguments in the discussion. I suggest careful rewording to make arguments more clear.

6. Is the description of experiments and calculations sufficiently complete and precise to allow their reproduction by fellow scientists (traceability of results)? Yes, for some of the methods, but not for others. See additional comments on the manuscript.

7. Do the authors give proper credit to related work and clearly indicate their own new/original contribution? Yes, citations seem appropriate.

8. Does the title clearly reflect the contents of the paper? Yes, the title seems appropriate.

9. Does the abstract provide a concise and complete summary? Overall the abstract is okay, but somewhat misrepresents the results for precipitation reconstructions and streamflow reconstructions. See additional comments.

10. Is the overall presentation well structured and clear? Not always. See additional comments on manuscript.

11. Is the language fluent and precise? In general, the language is appropriate, although some wording is a bit awkward or unusual. In addition, I had a lot of difficulty following the discussion and suggest the authors reword most of the text there to be more clear about what they are arguing. See additional comments on manuscript.

12. Are the mathematical formulae, symbols, abbreviations, and units correctly defined and used? The formulae provided seem adequately described. However, I would encourage including some equations regarding the rainfall-runoff model. In addition, the description of some of the parameters in Equation (4) do not seem correct (see additional comments on manuscript).

13. Should any parts of the paper (text, formulae, figures, tables) be clarified, reduced, combined, or eliminated? In general the organization is appropriate, but there are a few places of repetition, and some clarification of figures or tables might be helpful (see additional comments on manuscript). The colors on the figures are hard to discern, especially for the light grey and grey lines.

14. Are the number and quality of references appropriate? Yes.

15. Is the amount and quality of supplementary material appropriate? I did not review any supplementary material.

Overall, the questions addressed by the modeling effort are interesting and the results presented are also interesting. However, I do not feel enough information has

been provided to substantiate the findings of the paper due to the lack of detail on the rainfall-runoff modeling. The authors refer to several citations about the model, but the application of the model to this study should be justified. To do this, information on how the model was calibrated needs to be described to show that such calibration was appropriate for the current use. Performance metrics of the calibration should be included. In addition, the model itself needs to be described regarding what inputs are needed, what the "4 and 2 free parameters to calibrate" are (that wording was very confusing to me; line 429). There should also be a description of what those calibrated parameters were and whether their values are appropriate. Their influence on the model results for the study described in this manuscript would also be helpful, given that streamflow reconstruction with the model had discrepancies.

I suggest that the authors need to make clearer the inputs needed for the reconstructed streamflows – I assumed it was time series of air temperatures and precipitation only, but that never clearly stated. The timestep necessary for these inputs also should be clear.

This relates to another thing that was unclear to me regarding why the authors used daily data if all of the comparisons/results shown were monthly. I am guessing the reason is possibly because the rainfall-runoff model only operated at the monthly timestep (relates to the lack of detail on the rainfall-runoff model). Alternatively, perhaps the reservoir operations would like daily data and hence, the approach needs to produce daily data. If this latter is the case, then the authors should present daily results and model performance as well, even if they do not perform as strongly as the monthly summaries of results. Regardless, there needs to be some explanation regarding why daily inputs are needed, but only monthly and annual results are reported.

I also had some difficulty following the terms used by the authors. This may be because I am not an atmospheric scientist and if the journal feels that its audience is most likely to follow the terminology used then these comments may not be valid. In particular, I was not familiar with "geopotential height," which therefore made discussion of one of

the primary datasets used for the reconstructions to be very difficult for me to follow. I recommend if the audience for this article is likely to be interdisciplinary, that the authors provide more description of what geopotential height is and how that relates to the data they used in their study. Also, the authors use "reconstitute" or "reconstitution" quite a bit in the manuscript. I think a more appropriate word is "reconstruct" or "reconstruction." The meaning of "reconstitute" is different from "reconstruct" and I think it is inappropriate here.

Additional comments are as follows (note: line numbers refer to manuscript-version1):

1. Abstract: Suggest rewording line 9 "to compare the obtained streamflow series" to something like "to compare streamflow series obtained with the new method" to be more clear (but also, compare to what?)

2. Line 58: The colon (:) after "Canada" seems inappropriate. I suggest just starting a new sentence with "The length (number of years)..."

3. Line 59: What is "(cQ)2"? Is this an abbreviation for something? If it is a publically available database, should a website be given?

4. Line 87: Suggest changing "consisting in cal-" to "consisting of cal-"

5. Line 143: Is 15,240 megawatts for the whole complex or just for Caniapiscau Reservoir?

6. Section 2.1.1: I am unfamiliar with geopotential height reanalysis and a couple of sentences here to define the approach would be useful.

7. Lines 195-196: I did not understand what the "5 first" were that were extracted – what determines what are first and last in the 56 members?

8. Lines 203-204: Keep the greater than sign (>) with the numbers (i.e., >100)

9. Lines 242-244: This is a fragment sentence – please reword

10. Lines 247-248: What is meant by "A daily catchment series" – do you mean a series of air temperatures for the catchment of Caniapiscau reservoir?

11. Line 255: Change "is coming" to "comes"

12. Line 258: change "system" to "systems"

13. Lines 258-259: Why is the La Grande system one of the most important hydropower systems in the world?

14. Line 265: Should "abound" be "around"?

15. Line 314: What do pressure fields have to do with analogue days?

16. Line 314: change "fields" to "field"

17. Section 3.1.1: The authors made a good attempt to explain this complicated process of finding analogue days, and Table 1 was helpful. More detail on the Teweles and Wobus (1954) distance is needed – I was not familiar with it, so lines 359-362 were not helpful in describing how the ranking was done (I also suggest avoiding such colloquial phrasing as "thanks to" to be more clear). As I interpreted by reading between the lines, it looks like 20 time series were created for M1, 20 time series were created for M2, and so on. If so, could that also be explicitly stated?

18. Line 404: I think a closing parenthesis is missing for "T(dk)"

19. Line 410: Delete "In conclusion," – the paper is not finished yet.

20. Lines 414-419: I suggest deleting these two sentences as they are repetitive with statements in Section 3.1.2.

21. Section 3.2: Please see previous comments about needing more detail on the rainfall-runoff model.

22. Lines 439-444: Description of the Kuentz et al. (2013) study belongs more in the discussion where the authors could compare their results with those of the previous

(similar) study.

23. Line 455: State what is a good value versus a bad value for KGE (i.e., is 1 best?)

24. Lines 458-462: Wouldn't all values of beta be positive, thus what type of values would indicate an overestimation (perhaps values >1)?

25. Lines 463-468: Wouldn't all values of alpha be positive, thus what type of values would indicate an overdispersion?

26. Lines 469-473: It probably would be helpful to indicate what value is a better result (i.e., 1 is a perfect correlation)

27. Line 496: delete "of" before "yearly"

28. Lines 513-522: Isn't the ANA with the line over it representing the average of the five 20CR members? If so, isn't it expected that it would have less variability than the individual reconstructions? I do suggest that a definition of the terms with the lines over them (5 ANA with line over it and 5 ANATEM with line over it) be given in the text and in the figure captions

29. Lines 523-540: I think that the use of the term "time step" is incorrect here unless the modeling was truly done at different time steps (which should be clearly explained if so). Otherwise, "period" or "resolution" would be more appropriate.

30. Line 540: I suggest using "as expected" rather than "logically" or else explain what you are considering as logical.

31. Section 4.1.2: Is the TEM series referred to here the BEST series?

32. Section 4.2: I was not clear about how this section was providing different information than Section 4.3.1. Perhaps those two sections could be combined?

33. Lines 635-644: Is this paragraph and Figure 7 about output from CemaNeige model? If so, please state so.

34. Lines 635-644: Why is there a focus on May values? Is this an important month or is it the month with the best fits?

35. Section 4.3.2: Are the reconstructions described here using CemaNeige model?

36. Lines 697-703: How did you determine that the 1950-60 period is an "average period" – was there a statistical analysis done to determine this, or are you arbitrarily deciding it is so?

37. Section 5: I would like to see a discussion of the parameters and limitations of the rainfall-runoff model. Were assumptions made with the rainfall-runoff model reasonable for this application?

38. Line 779: change "representing" to "represent"

39. Lines 799-812: I do not follow the text here. What limited performances are being referred to? What did Kuentz et al. (2015) highlight? How does the work have a perspective of finding an additional series? Is that done and described (I don't think so, but I couldn't really tell what was being stated here)? Please elaborate more on how variables like relative humidity, precipitable water content (what is this?), and local pressure measurements would be used. Would they be used in the rainfall-runoff model? Would they be used to reconstruct precipitation or air temperature? Where would these variables come from? Are they something that you can get from geopotential height? When reconstructing into the past, how you do you estimate these variables? Or are you intending to just reconstruct back through the observational record rather than for centuries as would be done with paleoreconstructions using tree-ring data?

40. Lines 813-823: Although the sensitivity analyses results are not shown, it would be useful to know what variables or approaches were sensitive. I did not follow the last sentence – was this lack of uncertainty shown in the results, and if so, can the authors point the reader to what they are referring to?

41. Line 825: Should "model" be added after "rainfall-runoff"?

42. Lines 824-839: I do not follow what this paragraph is arguing. How (and why) would the parameter set change in changing climate? What parameter set are you talking about – the ones for the rainfall-runoff model, or perhaps the ones for Equation (1)? Please reword the entire paragraph to be more clear.

43. Line 859: change "focusing" to "focus"

44. Figures 3, 4, 5, 6, 8, 9: I have a very difficult time making out the 5*20 ANATEM or 5*20 ANA data in these figures. I cannot distinguish 5*ANA from 5*20 ANATEM in Figure 3. I suggest the authors consider using some different colors for these lines or symbols.

45. Figure 7: Suggest moving "(a)" before "mean annual streamflow" and "(b)" before "May monthly"

46. Figure 9: Should the reference to Nicault et al. (2014) in the caption actually be to Boucher et al. (2011)?

---

## Referee Comment (RC2) · Anonymous Referee #2 · 8 Apr 2016

Dear Editor, Please find here below my review of the paper cp-2016-5:

Streamflow variability over 1881-2011 period in northern Quebec: comparison of hydrological reconstructions based on tree rings and on geopotential height field reanalysis By P. Brigode, F. Brissette, A. Nicault, L. Perreault, A. Kuentz, T. Mathevet, and J. Gailhard

This paper presents the reconstruction of the streamflow over a past period of time for a catchment in northern Quebec. It is based on analogue method using reanalysis of the past geopotential. This method allows to provide precipitations per year based on

similar synoptic situations. These rainfall are later transformed in runoff by a second model. Comparing the results with the tree-ring model and spring floods analysis, the authors concluded that the results are promising.

1. Does the paper address relevant scientific questions within the scope of CP? Yes 2. Does the paper present novel concepts, ideas, tools, or data? Yes partly, it uses different know concepts but present an original approach. 3. Are substantial conclusions reached? Yes it shows the potential of the proposed method. 4. Are the scientific methods and assumptions valid and clearly outlined? In some aspect can be a bit improved some parameters are missing 5. Are the results sufficient to support the interpretations and conclusions? Yes 6. Is the description of experiments and calculations sufficiently complete and precise to allow their reproduction by fellow scientists (traceability of results)? Clearly missing some inputs to reproduce the results 7. Do the authors give proper credit to related work and clearly indicate their own new/original contribution? Yes 8. Does the title clearly reflect the contents of the paper? Yes 9. Does the abstract provide a concise and complete summary? Yes 10. Is the overall presentation well structured and clear? Yes 11. Is the language fluent and precise? Yes 12. Are mathematical formulae, symbols, abbreviations, and units correctly defined and used? Not really a lot of equations... 13. Should any parts of the paper (text, formulae, figures, tables) be clarified, reduced, combined, or eliminated? No, but some improvement are possible see general comments 14. Are the number and quality of references appropriate? Yes 15. Is the amount and quality of supplementary material appropriate? It can maybe be used, see general comments.

General comments

The paper is well written and is in a form very similar to other paper on the paleo climate text. It is rather long, but using several methods, this is necessary to present everything. Nevertheless, there is not always justification of the choices. For instance the choice of the zone used for the geopotential is not justified. And some parameters for the different models are not explicit. If possible it would be nice to integrate them

in a way, but I know it is an issue because the paper will be longer. Because it is a long paper using several concepts, I would recommend to the author to summarize in a flow-chart figure each step of their methodology to reach streamflow. It would make it easier for the reader to follow the whole text. If the author can take this remarks into account, the paper will be nearly ready for publication.

Specific comments:

A few specific comments: Fig 1: I do not recognize the catchment on figure 1b? why? Page 3 line 4 add reference after "dendrohydrology". Legend figure 4: add "for" 1950?? Page 15 line 1: blank after the dot. Figure 9: I do not understand tree ring reference to Nicault and Boucher in b?

---

## Author Comment (AC1) · 5 May 2016

**Streamflow variability over 1881–2011 period in northern Quebec: Comparison of hydrological reconstructions based on tree rings and on geopotential height field reanalysis**
**Paper *cp-2016-5**

by Brigode, P.; Brissette, F.; Nicault, A.; Perreault, L.; Kuentz, A.; Mathevet, T. & Gailhard, J.

**Answer to the referee comments**

Comments and suggestions made by the two referees are gratefully acknowledged. We intend to modify the text in response to the main criticisms. In the following, we list the referee comments (in *italic and blue*) and we provide specific responses to these comments (in black).

Furthermore, several modifications will be made in the manuscript related to minor errors detected:

- **Wrong co-authors affiliations:**

  We will correct the wrong co-authors affiliations.

  P. Brigode[1,2,*], F. Brissette[1], A. Nicault[3], L. Perreault[4], A. Kuentz[5], T. Mathevet[6], and J. Gailhard[6]

  [1] Ecole de Technologie Supérieure de Montréal, Montréal, Canada

  [2] Ouranos, Montréal, Canada

  [3] ECCOREV, Aix-en-Provence, France

  [4] IREQ, Varennes, Canada

  [5] SMHI, Norrkoping, Sweden

  [6] DTG, DMM, Electricité de France, Grenoble, France

  [*]Now in: Université de Nice Sophia Antipolis, CNRS, IRD, OCA, Géoazur UMR 7329

- **Wrong reference of the work of Way and Viau (2014) in Labrador as being in New-Brunswick**

  We will correct this mistake.

**1 REFEREE #1**

**1.1 General comments**

*Overall, the questions addressed by the modeling effort are interesting and the results presented are also interesting. However, I do not feel enough information has been provided to substantiate the findings of the paper due to the lack of detail on the rainfall-runoff modeling. The authors refer to several citations about the model, but the application of the model to this study should be justified. To do this, information on how the model was calibrated needs to be described to show that such calibration was appropriate for the current use. Performance metrics of the calibration should be included. In addition, the model itself needs to be described regarding what inputs are needed, what the "4 and 2 free parameters to calibrate" are (that wording was very confusing to me; line 429). There should also be a description of what those calibrated parameters were and whether their values are appropriate. Their influence on the model results for the study described in this manuscript would also be helpful, given that streamflow reconstruction with the model had discrepancies.*

We agree with the referee #1 that the description of the rainfall-runoff model was lacking important information in the present form. A complete description of the GR4J model and its snowmelt routine CemaNeige will be added, with a focus on the inputs needed and on the timestep of the model. A table will also be added, giving a description of each of the six calibrated parameters, their unit and their final calibrated values. We will clearly define what a "free parameter to calibrate" is (terminology classically used in the rainfall-runoff modeling community).

Although there is an entire result subsection devoted to the rainfall-runoff model calibration performances (*4.2 Rainfall-runoff model calibration performances*), we will add the calibration metric values obtained after calibration (Kling and Gupta Efficiency score (Gupta *et al.*, 2009) and its three components).

Finally, quantifying the influence of each rainfall-runoff model parameter on the final streamflow reconstruction is out of the scope of this paper and is definitively an open question (and thus an interesting perspectives of this work). Here, the idea was to apply a classical rainfall-runoff model calibration strategy and then used the obtained parameter values in order to have a model able to transform an ensemble of daily climatic series into an ensemble of daily streamflow series. Nevertheless, our expert (and thus biased) judgement, as hydrologists, is that the rainfall-runoff transformation is not a "significant issue" on this catchment, mainly due to its topographic (topography relatively flat) and hydro-climatic context (catchment hydrology strongly influenced by snowmelt, with slow flow dynamics and none sudden events) and its (very) large size.

*I suggest that the authors need to make clearer the inputs needed for the reconstructed streamflows – I assumed it was time series of air temperatures and precipitation only, but that never clearly stated. The timestep necessary for these inputs also should be clear.*

We agree that the timestep of the model was not clearly stated. The climatic reconstruction described in this paper is done at the daily timestep and the rainfall-runoff model used is also operating at the daily timestep. Thus, input and output series are all at the daily timestep. It will be clearly stated in the manuscript.

*This relates to another thing that was unclear to me regarding why the authors used daily data if all of the comparisons/results shown were monthly. I am guessing the reason is possibly because the rainfall-runoff model only operated at the monthly timestep (relates to the lack of detail on the rainfall-runoff model). Alternatively, perhaps the reservoir operations would like daily data and hence, the approach needs to produce daily data. If this latter is the case, then the authors should present daily results and model performance as well, even if they do not perform as strongly as the monthly summaries of results. Regardless, there needs to be some explanation regarding why daily inputs are needed, but only monthly and annual results are reported.*

Monthly and annual values are showed in the paper because of the main goal, which is to compare the new streamflow reconstruction with two other reconstructions (using tree-rings) available at the annual timestep. Nevertheless, as detailed in the previous answers, outputs of the reconstruction methodology are available at the daily timestep. We will evaluated the performance of the climatic and hydrologic performances at the daily timestep and present it alongside the monthly and annual performances (in figures 4 and 7).

*I also had some difficulty following the terms used by the authors. This may be because I am not an atmospheric scientist and if the journal feels that its audience is most likely to follow the terminology used then these comments may not be valid. In particular, I was not familiar with "geopotential height," which therefore made discussion of one of the primary datasets used for the reconstructions to be very difficult for me to follow. I recommend if the audience for this article is likely to be interdisciplinary, that the authors provide more description of what geopotential height is and how that relates to the data they used in their study. Also, the authors use "reconstitute" or "reconstitution" quite a bit in the manuscript. I think a more appropriate word is "reconstruct" or "reconstruction." The meaning of "reconstitute" is different from "reconstruct" and I think it is inappropriate here.*

Geopotential height fields will be clearly defined in the manuscript, based on the NOAA's National Weather Service Glossary. A geopotential height is the height above sea level of a given pressure level. For example, if a station reports that the 500 [hPa] height at its location is 5600 [meters], it means that the level of the atmosphere over that station at which the atmospheric pressure is 500 [hPa] is 5600 [meters] above sea level. Note that for pressure levels close from sea level pressure (typically 1000 [hPa]), the geopotential height could be negative.

Also, we will only use the words "reconstruct" and "reconstruction" in the manuscript.

**1.2   Specific comments**

*1. Abstract: Suggest rewording line 9 "to compare the obtained streamflow series" to something like "to compare streamflow series obtained with the new method" to be more clear (but also, compare to what?)*

Agreed, and we will explicitly state that we compare the streamflow series reconstructed in this article with two streamflow series obtained with tree ring data by other authors.

*2. Line 58: The colon (:) after "Canada" seems inappropriate. I suggest just starting a new sentence with "The length (number of years): : :"*

Agreed.

*3. Line 59: What is "(cQ)2"? Is this an abbreviation for something? If it is a publically available database, should a website be given?*

(cQ)² is the abbreviation for "*Impact des Changements Climatiques sur l'hydrologie (Q) au Québec*". The cQ2 database is not publically available.

*4. Line 87: Suggest changing "consisting in cal-" to "consisting of cal-"*

Agreed.

*5. Line 143: Is 15,240 megawatts for the whole complex or just for Caniapiscau Reservoir?*

It is for the whole complex. The revised total installed capacity is in fact 17 418 megawatts (to be corrected in the manuscript). The installed capacity for Brisay power plant at Caniapiscau is 469 megawatts.

*6. Section 2.1.1: I am unfamiliar with geopotential height reanalysis and a couple of sentences here to define the approach would be useful.*

We will introduce this sub-section by defining what a geopotential height reanalysis is and how it is generated (see answer to the general comments).

*7. Lines 195-196: I did not understand what the "5 first" were that were extracted –what determines what are first and last in the 56 members?*

We agree with the referee #1 that this sentence is unclear, and we will rephrase it. Since each member is equiprobable, selecting the members 1 to 5 (i.e. the "5 first") is equivalent to randomly selecting 5 members out of the 56 members available.

*8. Lines 203-204: Keep the greater than sign (>) with the numbers (i.e., >100)*

Agreed, we will used the sign here and in other equivalent sentences in the manuscript.

*9. Lines 242-244: This is a fragment sentence – please reword*

Yes, few words are missing in this sentence and we will reword it.

*10. Lines 247-248: What is meant by "A daily catchment series" – do you mean a series of air temperatures for the catchment of Caniapiscau reservoir?*

Yes, we meant that we used one and only daily series of air temperature for the entire catchment. We will rephrase this sentence in the manuscript.

*11. Line 255: Change "is coming" to "comes"*

Agreed.

*12. Line 258: change "system" to "systems"*

Agreed.

*13. Lines 258-259: Why is the La Grande system one of the most important hydropower systems in the world?*

The Three Gorges Dam is the most important hydropower system is the world with a total installed capacity of around 22 000 megawatts, the La Grande system has an installed capacity of 17 418 megawatts and is thus one of the most important hydropower system in the world. The Brisay power plant (at Caniapiscau) is ranked as the 9[th] with an installed capacity of around 500 megawatts.

*14. Line 265: Should "abound" be "around"?*

Yes, we will change this in the manuscript.

*15. Line 314: What do pressure fields have to do with analogue days?*

The term "pressure fields" is used here to describe the "geopotential height fields" (see answer to the general comments and to the specific point #6) which are used to find meteorological analogy between days: days with similar geopotential height fields are assumed to be meteorologically "analogue" and thus to produce similar temperature and precipitation pattern over a given region. We will rephrase this sentence in order to be clearer.

*16. Line 314: change "fields" to "field"*

Agreed.

*17. Section 3.1.1: The authors made a good attempt to explain this complicated process of finding analogue days, and Table 1 was helpful. More detail on the Teweles and Wobus (1954) distance is needed – I was not familiar with it, so lines 359-362 were not helpful in describing how the ranking was done (I also suggest avoiding such colloquial phrasing as "thanks to" to be more clear). As I interpreted by reading between the lines, it looks like 20 time series were created for M1, 20 time series were created for M2, and so on. If so, could that also be explicitly stated?*

We will add some details on how the Teweles and Wobus (1954) is calculated (the formula and an example of the calculation). The "thanks to" will be deleted. Finally, we will explicitly state that 20 time series are created for each considered members.

*18. Line 404: I think a closing parenthesis is missing for "T(dk)"*

Yes, we will add a closing parenthesis.

*19. Line 410: Delete "In conclusion," – the paper is not finished yet.*

Agreed.

*20. Lines 414-419: I suggest deleting these two sentences as they are repetitive with statements in Section 3.1.2.*

Agreed.

*21. Section 3.2: Please see previous comments about needing more detail on the rainfall-runoff model.*

Information about the rainfall-runoff model will be added in this section (see answer to general comment).

*22. Lines 439-444: Description of the Kuentz et al. (2013) study belongs more in the discussion where the authors could compare their results with those of the previous (similar) study.*

We will move this sentence to the discussion section.

*23. Line 455: State what is a good value versus a bad value for KGE (i.e., is 1 best?)*

We will explicitly state that a perfect KGE value is 1.

*24. Lines 458-462: Wouldn't all values of beta be positive, thus what type of values would indicate an overestimation (perhaps values >1)?*

The referee #1 is right, all beta values are positive and values greater than 1 indicate an overestimation while values lower than 1 indicate an underestimation. We will correct this mistake in the manuscript.

*25. Lines 463-468: Wouldn't all values of alpha be positive, thus what type of values would indicate an overdispersion?*

The referee #1 is right, all alpha values are positive and values greater than 1 indicate an overdispersion. We will correct this mistake in the manuscript.

*26. Lines 469-473: It probably would be helpful to indicate what value is a better result (i.e., 1 is a perfect correlation)*

Agreed.

*27. Line 496: delete "of" before "yearly"*

Agreed.

*28. Lines 513-522: Isn't the ANA with the line over it representing the average of the five 20CR members? If so, isn't it expected that it would have less variability than the individual reconstructions? I do suggest that a definition of the terms with the lines over them (5 ANA with line over it and 5 ANATEM with line over it) be given in the text and in the figure captions*

We agree with the referee #1: these lines and associated terms will be clearly defined in the text and in the figure captions, to avoid any confusion.

*29. Lines 523-540: I think that the use of the term "time step" is incorrect here unless the modeling was truly done at different time steps (which should be clearly explained if so). Otherwise, "period" or "resolution" would be more appropriate.*

Agreed, we will then use the "resolution" term.

*30. Line 540: I suggest using "as expected" rather than "logically" or else explain what you are considering as logical.*

We will use "as expected" rather than "logically".

*31. Section 4.1.2: Is the TEM series referred to here the BEST series?*

Yes, TEM referred here (and after) to the BEST series. In order to avoid any confusion, we will change here (and after) TEM to BEST.

*32. Section 4.2: I was not clear about how this section was providing different information than Section 4.3.1. Perhaps those two sections could be combined?*

These two sections are providing different information since the first one (section 4.2) details the rainfall-runoff model calibration performances (i.e. using observed air temperature and precipitation daily series for reproducing daily observed streamflow series), while the second one is giving detail on the ability of the reconstruction to reproduce observed streamflow (i.e. using reconstructed air temperature and precipitation series for reconstructing observed streamflow series).

We will consider to change the order of these subsections in the manuscript, by presenting them in this new order:

    4.1 Rainfall-runoff model calibration performances (1963-1979);
    4.2 Climatic reconstructions (1951-2010 and 1880-2011);
    4.3 Streamflow reconstructions (1962-2011 and 1881-2011);

*33. Lines 635-644: Is this paragraph and Figure 7 about output from CemaNeige model? If so, please state so.*

All the rainfall-runoff model outputs presented in the manuscript have been produced by using both GR4J rainfall-runoff model and its snowmelt routine CemaNeige. We will explicitly state so in the manuscript.

*34. Lines 635-644: Why is there a focus on May values? Is this an important month or is it the month with the best fits?*

There is a focus on the May values because Boucher *et al.* (2011) produced a May streamflow reconstruction, using both continuous series (tree ring minimal density measurements) and discrete series (with ice-scars due to ice abrasion during floods). This month is particularly important in this catchment since it is a month with a large increase of the streamflow and with the observation of the spring flood peak at the end of the month or in early June.

Nevertheless, we intend to change our definition of the spring flood in the manuscript, since it may produce some biases, for example for years for which the flood peak is observed in early June (and thus no more centered in May). Thus, we will produce several "annual spring flood series" from our daily streamflow series (e.g. mean May streamflow values, mean June streamflow values, mean May-June streamflow values, maximum of a moving 30-day window over the May-June period, etc.), and compare these series with the tree-ring reconstructions.

*35. Section 4.3.2: Are the reconstructions described here using CemaNeige model?*

Yes, see answer to the specific comment #33.

*36. Lines 697-703: How did you determine that the 1950-60 period is an "average period" – was there a statistical analysis done to determine this, or are you arbitrarily deciding it is so?*

The term "average" is arbitrary in this context, and is used here since the average of the May streamflow reconstructed using the tree-ring over this decade (1950-1960) is close from the overall May streamflow average value (1881-1980). We will change this descriptive term in the manuscript.

*37. Section 5: I would like to see a discussion of the parameters and limitations of the rainfall-runoff model. Were assumptions made with the rainfall-runoff model reasonable for this application?*

We will add a discussion about the rainfall-runoff transformation in this section, arguing that the assumptions made are reasonable regarding the performances obtained by the rainfall-runoff model over the calibration period (presented in the results section).

*38. Line 779: change "representing" to "represent"*

Agreed.

*39. Lines 799-812: I do not follow the text here. What limited performances are being referred to? What did Kuentz et al. (2015) highlight? How does the work have a perspective of finding an additional series? Is that done and described (I don't think so, but I couldn't really tell what was being stated here)? Please elaborate more on how variables like relative humidity, precipitable water content (what is this?), and local pressure measurements would be used. Would they be used in the rainfall-runoff model? Would they be used to reconstruct precipitation or air temperature? Where would these variables come from? Are they something that you can get from geopotential height? When reconstructing into the past, how you do you estimate these variables? Or are you intending to just reconstruct back through the observational record rather than for centuries as would be done with paleoreconstructions using tree-ring data?*

The limited performances referred here are the inability of the ANA approach to reproduce the long-term trend of climatic series (here temperature and precipitation), as already pointed out by Kuentz *et al.* (2015). Unfortunately, none long precipitation and temperature series are available in the studied region. The perspectives are thus to improve the current methodology and particularly testing variables available through the reanalysis for the analogy. Several authors used variables such as air temperature, vertical velocity and humidity at different atmospheric levels (variables produced by the

20CR reanalysis and thus available from 1851 to 2011) to find analogue dates and finally reconstruct daily air temperature and precipitation series. Trying to use such variables for the reconstruction and compare the obtained performances with and without these additional variables is an interesting perspective.

*40. Lines 813-823: Although the sensitivity analyses results are not shown, it would be useful to know what variables or approaches were sensitive. I did not follow the last sentence – was this lack of uncertainty shown in the results, and if so, can the authors point the reader to what they are referring to?*
*(This comment is found also in the general comment of the Referee #2).*
Several results of this sensitivity analysis (e.g. the spatial domain considered for the analogy) will be presented in a new Appendix part added to the manuscript. The last sentence was: "*Interestingly, the uncertainty due to the use of five members of the 20CR reanalysis appears to be limited, and even null from 1940 onward*". Yes, this "lack of uncertainty" is shown in results, see for example the Figure 8: it is impossible to distinguish the 5 ANATEM average series after 1940, highlighting that considering 5 different members of the 20CR reanalysis has a negligible impact on the reconstruction of the mean annual streamflow. We will explicitly point the reader to these figures in the manuscript.

*41. Line 825: Should "model" be added after "rainfall-runoff"?*
Yes, we will add "model" after "rainfall-runoff".

*42. Lines 824-839: I do not follow what this paragraph is arguing. How (and why) would the parameter set change in changing climate? What parameter set are you talking about – the ones for the rainfall-runoff model, or perhaps the ones for Equation (1)? Please reword the entire paragraph to be more clear.*
This paragraph is intend to reminding and discussing the assumptions made when using a (calibrated) rainfall-runoff model over a climatically-contrasted and long period of time. We thus talk about the parameter set of the rainfall-runoff model, obtained after a calibration over a short period (here 17 years). Numerous authors thus highlighted that calibrated parameter sets are dependent on the climate of the calibration period and that the rainfall-runoff models show limited performances when applied over periods that are climatically contrasted regarding to the climate of the calibration period. It is clearly out of the scope of this paper to quantify the sensitivity of the streamflow reconstruction to these "stationary" assumptions, but it is an interesting perspective of this work. We will reword this paragraph to be clear.

*43. Line 859: change "focusing" to "focus"*
Agreed.

*44. Figures 3, 4, 5, 6, 8, 9: I have a very difficult time making out the 5*20 ANATEM or 5*20 ANA data in these figures. I cannot distinguish 5*ANA from 5*20 ANATEM in Figure 3. I suggest the authors consider using some different colors for these lines or symbols.*
We agree that several lines are impossible to see or to distinguish on these figures. Even if this is a significant and interesting result (meaning that there is no dispersion between simulations or no difference between the observation and the simulation), we will change the colors and the point types and details the differences of the lines in the figure captions in order to distinguish the different simulations.

*45. Figure 7: Suggest moving "(a)" before "mean annual streamflow" and "(b)" before "May monthly"*
Agreed.

*46. Figure 9: Should the reference to Nicault et al. (2014) in the caption actually be to Boucher et al. (2011)?*

Yes, we will correct this mistake.

**2    REFEREE #2**

**2.1    General comments**

*The paper is well written and is in a form very similar to other paper on the paleoclimate text. It is rather long, but using several methods, this is necessary to present everything. Nevertheless, there is not always justification of the choices. For instance the choice of the zone used for the geopotential is not justified. And some parameters for the different models are not explicit. If possible it would be nice to integrate them in a way, but I know it is an issue because the paper will be longer. Because it is a long paper using several concepts, I would recommend to the author to summarize in a flow-chart figure each step of their methodology to reach streamflow. It would make it easier for the reader to follow the whole text. If the author can take this remarks into account, the paper will be nearly ready for publication.*

The tests performed for choosing the spatial domain considered for the geopotential height field will be presented in a new Appendix part of the manuscript.

We will also add several paragraphs in order to fully describe the rainfall-runoff model and its snowmelt routine and how are calibrated the parameters (cf. answers to Referee #1).

Moreover, adding a flowchart summarizing the entire methodology will definitely improve the manuscript, we would like to thank Referee #2 for this suggestion. We produced a new figure that we will add to the manuscript in order to summarize the reconstruction methodology applied:

[Figure]

**2.2 Specific comments**

*Fig 1: I do not recognize the catchment on figure 1b? why?*
The studied catchment is one of the 211 cQ2 catchments is thus plotted in the Figure 1b, but is hidden by an intermediate sub-catchment. The Caniapiscau catchment will be highlighted in the manuscript with shading lines.

*Page 3 line 4 add reference after "dendrohydrology".*
We will add the reference to the review of Loaiciga *et al.* (1993).

*Legend figure 4: add "for" 1950??*
Agreed.

*Page 15 line 1: blank after the dot.*
We will add a space.

*Figure 9: I do not understand tree ring reference to Nicault and Boucher in b?*
The "tree-ring series" presented in the Figure 9 is from Boucher *et al.* (2011), we will thus correct the Figure 9 legend.

**3  REFERENCES**

Boucher, É., Ouarda, T.B.M.J., Bégin, Y., Nicault, A., 2011. Spring flood reconstruction from continuous and discrete tree ring series. Water Resour. Res. 47, W07516. doi:10.1029/2010WR010131

Gupta, H.V., Kling, H., Yilmaz, K.K., Martinez, G.F., 2009. Decomposition of the mean squared error and NSE performance criteria: Implications for improving hydrological modelling. Journal of Hydrology 377, 80–91. doi:16/j.jhydrol.2009.08.003

Kuentz, A., Mathevet, T., Gailhard, J., Hingray, B., 2015. Building long-term and high spatio-temporal resolution precipitation and air temperature reanalyses by mixing local observations and global atmospheric reanalyses: the ANATEM model. Hydrol. Earth Syst. Sci. 19, 2717–2736. doi:10.5194/hess-19-2717-2015

Loaiciga, H.A., Haston, L., Michaelsen, J., 1993. Dendrohydrology and long-term hydrologic phenomena. Reviews of Geophysics 31, 151–171.

Teweles, J., Wobus, H., 1954. Verification of prognosis charts. Bulletin of the American Meteorological Society 35, 455–463.

Way, R.G., Viau, A.E., 2014. Natural and forced air temperature variability in the Labrador region of Canada during the past century. Theor Appl Climatol 121, 413–424. doi:10.1007/s00704-014-1248-2

---

## Author Response (AR1)

*Streamflow variability over 1881–2011 period in northern Quebec:*

*Comparison of hydrological reconstructions based on tree rings and on geopotential height field reanalysis*

Paper *cp-2016-5*

by Brigode, P.; Brissette, F.; Nicault, A.; Perreault, L.; Kuentz, A.; Mathevet, T. & Gailhard, J.

**Answer to the referee comments and markep-up manuscript version**

Comments and suggestions made by the two referees are gratefully acknowledged. We modified the text in response to the main criticisms. In the following, we list the referee comments (in *italic and blue*), we provide specific responses to these comments (in black) and finally we present a marked-up manuscript version.

**1    REFEREE #1**

**1.1    General comments**

*Overall, the questions addressed by the modeling effort are interesting and the results presented are also interesting. However, I do not feel enough information has been provided to substantiate the findings of the paper due to the lack of detail on the rainfall-runoff modeling. The authors refer to several citations about the model, but the application of the model to this study should be justified. To do this, information on how the model was calibrated needs to be described to show that such calibration was appropriate for the current use. Performance metrics of the calibration should be included. In addition, the model itself needs to be described regarding what inputs are needed, what the "4 and 2 free parameters to calibrate" are (that wording was very confusing to me; line 429). There should also be a description of what those calibrated parameters were and whether their values are appropriate. Their influence on the model results for the study described in this manuscript would also be helpful, given that streamflow reconstruction with the model had discrepancies.*

We agree with the referee #1 that the description of the rainfall-runoff model was lacking important information in the present form. A complete description of the GR4J model and its snowmelt routine CemaNeige has ben added in the section 3 (*Methodology*), with a focus on the inputs needed and on the timestep of the model:

*The structure of the CemaNeigeGR4J model is presented in the Figure 3. GR4J is based on two non-linear stores (production and routing stores) and a unit-hydrograph, while CemaNeige is a degree-day snow accounting routine, which divides the studied catchment into five elevation bands. CemaNeigeGR4J uses as inputs daily series of precipitation, minimal and maximal air temperatures and a daily potential evapotranspiration series, calculated using Oudin et al. (2005) formula, designed for rainfall-runoff modelling. CemaNeigeGR4J produces daily streamflow series.*

A table has also been added in this section (Table 1), giving a description of each of the six calibrated parameters, their unit and their final calibrated values.

Although there is an entire result subsection devoted to the rainfall-runoff model calibration performances (*4.1 Rainfall-runoff model calibration performances*), we added the calibration metric values obtained after calibration (Kling and Gupta Efficiency score (Gupta *et al.*, 2009), KGE = 0.93).

Finally, quantifying the influence of each rainfall-runoff model parameter on the final streamflow reconstruction is out of the scope of this paper and is definitely an open question (and thus an interesting perspectives of this work). Here, the idea was to apply a classical rainfall-runoff model calibration strategy and then used the obtained parameter values in order to have a model able to transform an ensemble of daily climatic series into an ensemble of daily streamflow series. Nevertheless, our expert (and thus biased) judgement, as hydrologists, is that the rainfall-runoff transformation is not a "significant issue" on this catchment, mainly due to its topographic (topography relatively flat) and hydro-climatic context (catchment hydrology strongly influenced by snowmelt, with slow flow dynamics and none sudden events) and its (very) large size.

*I suggest that the authors need to make clearer the inputs needed for the reconstructed streamflows – I assumed it was time series of air temperatures and precipitation only, but that never clearly stated. The timestep necessary for these inputs also should be clear.*

We agree that the timestep of the model was not clearly stated. The climatic reconstruction described in this paper is done at the daily timestep and the rainfall-runoff model used is also operating at the daily timestep. Thus, input and output series are all at the daily timestep. It is now clearly stated in the manuscript (see previous answer).

*This relates to another thing that was unclear to me regarding why the authors used daily data if all of the comparisons/results shown were monthly. I am guessing the reason is possibly because the rainfall-runoff model only operated at the monthly timestep (relates to the lack of detail on the rainfall-runoff model). Alternatively, perhaps the reservoir operations would like daily data and hence, the approach needs to produce daily data. If this latter is the case, then the authors should present daily results and model performance as well, even if they do not perform as strongly as the monthly summaries of results. Regardless, there needs to be some explanation regarding why daily inputs are needed, but only monthly and annual results are reported.*

Monthly and annual values are showed in the paper because of the main article goal, which is to compare the new streamflow reconstruction with two other reconstructions (using tree-rings) available at the annual resolution. Nevertheless, as detailed in the previous answers, outputs of the reconstruction methodology are available at the daily resolution. We now evaluate the performance of the climatic reconstruction at the daily timestep and present it alongside the monthly and annual performances (see for example Figure 6).

*I also had some difficulty following the terms used by the authors. This may be because I am not an atmospheric scientist and if the journal feels that its audience is most likely to follow the terminology used then these comments may not be valid. In particular, I was not familiar with "geopotential height," which therefore made discussion of one of the primary datasets used for the reconstructions to be very difficult for me to follow. I recommend if the audience for this article is likely to be interdisciplinary, that the authors provide more description of what geopotential height is and how that relates to the data they used in their study. Also, the authors use "reconstitute" or "reconstitution" quite a bit in the manuscript. I think a more appropriate word is "reconstruct" or "reconstruction." The meaning of "reconstitute" is different from "reconstruct" and I think it is inappropriate here.*

Geopotential height fields is now clearly defined in the manuscript, in section 2.1 (*Datasets used for the climatic reconstructions*):

*A geopotential height is the height above sea level of a given pressure level. For example, if a station reports that the 500 hPa height at its location is 5600 meters, it means that the level of the atmosphere over that station at which the atmospheric pressure is 500 hPa is 5600 meters above sea level (example from the NOAA's National Weather Service). Note that for pressure levels close to sea level (typically 1000 hPa), the geopotential height can sometimes be negative. The analysis of geopotential height*

*fields over a given domain describes the spatial distribution of high/low pressure systems upon which similarity in between days can be measured.*

Also, we now only use the words "reconstruct" and "reconstruction" in the manuscript.

**1.2 Specific comments**

*1. Abstract: Suggest rewording line 9 "to compare the obtained streamflow series" to something like "to compare streamflow series obtained with the new method" to be more clear (but also, compare to what?)*

Agreed, and we now explicitly state that we compare the streamflow series reconstructed in this article with two streamflow series obtained with tree ring data by other authors:

*In this paper, we applied a new hydro-climatic reconstruction method on the Caniapiscau Reservoir and compare the obtained streamflow time series against time series derived from dendrohydrology by other authors on the same catchment and study the natural streamflow variability over the 1881-2011 period in that region.*

*2. Line 58: The colon (:) after "Canada" seems inappropriate. I suggest just starting a new sentence with "The length (number of years): : :"*

Agreed.

*3. Line 59: What is "(cQ)2"? Is this an abbreviation for something? If it is a publically available database, should a website be given?*

(cQ)² is the abbreviation for "*Impact des Changements Climatiques sur l'hydrologie (Q) au Québec*". The cQ2 database is not publically available.

*4. Line 87: Suggest changing "consisting in cal-" to "consisting of cal-"*

Agreed.

*5. Line 143: Is 15,240 megawatts for the whole complex or just for Caniapiscau Reservoir?*

It is for the whole complex. The revised total installed capacity is in fact 17 418 megawatts (now corrected in the manuscript). The installed capacity for Brisay power plant at Caniapiscau is 469 megawatts.

*6. Section 2.1.1: I am unfamiliar with geopotential height reanalysis and a couple of sentences here to define the approach would be useful.*

We now introduce this sub-section by defining what a geopotential height reanalysis is and how it is generated (see answer to the general comments).

*7. Lines 195-196: I did not understand what the "5 first" were that were extracted –what determines what are first and last in the 56 members?*

We agree with the referee #1 that this sentence was unclear, and we thus rephrased it. Since each member is equiprobable, selecting the members 1 to 5 (i.e. the "5 first") is equivalent to randomly selecting 5 members out of the 56 members available.

*Of the 56 ensemble members constituting the 20CR reanalysis, the members 1 to 5 were extracted and used over this region (see section 3.1.1***Erreur ! Source du renvoi introuvable.** *for more details).*

*8. Lines 203-204: Keep the greater than sign (>) with the numbers (i.e., >100)*

Agreed, we now use the sign here and in other equivalent sentences in the manuscript.

*9. Lines 242-244: This is a fragment sentence – please reword*

Yes, few words were missing in this sentence and we thus reworded it:

*For the air temperature, the Berkeley Earth Surface Temperature (hereafter denoted as BEST) analysis has been used, taken from the http://berkeleyearth.org/ Web site (Rohde et al., 2013).*

*10. Lines 247-248: What is meant by "A daily catchment series" – do you mean a series of air temperatures for the catchment of Caniapiscau reservoir?*

Yes, we meant that we used one and only daily series of air temperature for the entire catchment.

*11. Line 255: Change "is coming" to "comes"*

Agreed.

*12. Line 258: change "system" to "systems"*

Agreed.

*13. Lines 258-259: Why is the La Grande system one of the most important hydropower systems in the world?*

The Three Gorges Dam is the most important hydropower system is the world with a total installed capacity of around 22 000 megawatts, the La Grande system has an installed capacity of 17 418 megawatts and is thus one of the most important hydropower system in the world. The Brisay power plant (at Caniapiscau) is ranked as the 9th with an installed capacity of around 500 megawatts.

*14. Line 265: Should "abound" be "around"?*

Yes, we changed this in the manuscript.

*15. Line 314: What do pressure fields have to do with analogue days?*

The term "pressure fields" is used here to describe the "geopotential height fields" (see answer to the general comments and to the specific point #6) which are used to find meteorological analogy between days: days with similar geopotential height fields are assumed to be meteorologically "analogue" and thus to produce similar temperature and precipitation pattern over a given region. We rephrased this sentence in order to be clearer:

*The ANATEM method (Kuentz et al., 2015) is built on the combination of two approaches: (i) the ANA (which stands for "ANAlogue") approach, that aims to find, for a given day, a given number of analogue days, based on the similarity of synoptic circulation (Obled et al., 2002) and (ii) the TEM (which stands for "TEMoin", the French word for "witness") approach, which is a basic regression model that uses a continuous and long-term reference (the witness) climatic series to reconstruct past climate.*

*16. Line 314: change "fields" to "field"*

Agreed.

*17. Section 3.1.1: The authors made a good attempt to explain this complicated process of finding analogue days, and Table 1 was helpful. More detail on the Teweles and Wobus (1954) distance is*

*needed – I was not familiar with it, so lines 359-362 were not helpful in describing how the ranking was done (I also suggest avoiding such colloquial phrasing as "thanks to" to be more clear). As I interpreted by reading between the lines, it looks like 20 time series were created for M1, 20 time series were created for M2, and so on. If so, could that also be explicitly stated?*

We added some details on how the Teweles and Wobus (1954) distance is calculated (the formula and an example of the calculation) in a new Appendix part of the paper (Appendix B). The "thanks to" has been deleted. Finally, we now explicitly state that 20 time series are created for each considered members:

*For each day, the 100 climatic values are obtained based on the 20 "closest" analogue days for each of the 5 20CR members considered.*

*18. Line 404: I think a closing parenthesis is missing for "T(dk)"*

Yes, we added a closing parenthesis.

*19. Line 410: Delete "In conclusion," – the paper is not finished yet.*

Agreed.

*20. Lines 414-419: I suggest deleting these two sentences as they are repetitive with statements in Section 3.1.2.*

Agreed.

*21. Section 3.2: Please see previous comments about needing more detail on the rainfall-runoff model.*

Information about the rainfall-runoff model has been added in this section (see answer to general comment).

*22. Lines 439-444: Description of the Kuentz et al. (2013) study belongs more in the discussion where the authors could compare their results with those of the previous (similar) study.*

We moved this sentence to the discussion section.

*23. Line 455: State what is a good value versus a bad value for KGE (i.e., is 1 best?)*

We now explicitly state that a perfect KGE value is 1:

*The KGE criterion ranges between -∞ and 1 (perfect simulation).*

*24. Lines 458-462: Wouldn't all values of beta be positive, thus what type of values would indicate an overestimation (perhaps values >1)?*

The referee #1 is right, all beta values are positive and values greater than 1 indicate an overestimation while values lower than 1 indicate an underestimation. We corrected this mistake in the manuscript.

*25. Lines 463-468: Wouldn't all values of alpha be positive, thus what type of values would indicate an overdispersion?*

The referee #1 is right, all alpha values are positive and values greater than 1 indicate an overdispersion. We corrected this mistake in the manuscript.

*26. Lines 469-473: It probably would be helpful to indicate what value is a better result (i.e., 1 is a perfect correlation)*

Agreed.

*27. Line 496: delete "of" before "yearly"*

Agreed.

*28. Lines 513-522: Isn't the ANA with the line over it representing the average of the five 20CR members? If so, isn't it expected that it would have less variability than the individual reconstructions? I do suggest that a definition of the terms with the lines over them (5 ANA with line over it and 5 ANATEM with line over it) be given in the text and in the figure captions*

We agree with the referee #1: these lines and associated terms are clearly defined in the text (in section 3.5 *Comparison of reconstructed series against observations*) and are now distinguishable in the figure, to avoid any confusion:

*In order to compare the reconstructed streamflow time series against observations, the reconstructed ensembles were first aggregated: a daily series was generated for each of the five 20CR members considered by averaging the 20 daily series constituting each ensemble. The five daily mean series are denoted as $\overline{ANA}$ or $\overline{ANATEM}$, depending on the method used to produce them.*

*29. Lines 523-540: I think that the use of the term "time step" is incorrect here unless the modeling was truly done at different time steps (which should be clearly explained if so). Otherwise, "period" or "resolution" would be more appropriate.*

Agreed, we now use the "resolution" term in the entire manuscript.

*30. Line 540: I suggest using "as expected" rather than "logically" or else explain what you are considering as logical.*

We reworded this sentence:

*Averaging each ensemble of the considered 20CR members (blue points) results in better temporal correlations at the daily and yearly resolutions, but at the expense of lower variability reproduction performance.*

*31. Section 4.1.2: Is the TEM series referred to here the BEST series?*

Yes, TEM referred here (and after) to the BEST series. In order to avoid any confusion, we changed here (and after) TEM to BEST.

*32. Section 4.2: I was not clear about how this section was providing different information than Section 4.3.1. Perhaps those two sections could be combined?*

These two sections are providing different information since the first one (section 4.2) details the rainfall-runoff model calibration performances (i.e. using observed air temperature and precipitation daily series for reproducing daily observed streamflow series), while the second one is giving detail on the ability of the reconstruction to reproduce observed streamflow (i.e. using reconstructed air temperature and precipitation series for reconstructing observed streamflow series).

We changed the order of these subsections in the manuscript, by presenting them in this new order:

    *4.1 Rainfall-runoff model calibration performances (1963-1979);*

    *4.2 Climatic reconstructions (1951-2010 and 1880-2011);*

    *4.3 Streamflow reconstructions (1962-2011 and 1881-2011);*

*33. Lines 635-644: Is this paragraph and Figure 7 about output from CemaNeige model? If so, please state so.*

All the rainfall-runoff model outputs presented in the manuscript have been produced by using both GR4J rainfall-runoff model and its snowmelt routine CemaNeige. We now explicitly state so in the manuscript:

*All the rainfall-runoff model outputs presented in the manuscript have been produced at the daily resolution by using both GR4J rainfall-runoff model and its snowmelt routine CemaNeige.*

*34. Lines 635-644: Why is there a focus on May values? Is this an important month or is it the month with the best fits?*

There is a focus on the May values because Boucher *et al.* (2011) produced a May streamflow reconstruction, using both continuous series (tree ring minimal density measurements) and discrete series (with ice-scars due to ice abrasion during floods). This month is particularly important in this catchment since it is a month with a large increase of the streamflow and with the observation of the spring flood peak at the end of the month or in early June.

*35. Section 4.3.2: Are the reconstructions described here using CemaNeige model?*

Yes, see answer to the specific comment #33.

*36. Lines 697-703: How did you determine that the 1950-60 period is an "average period" – was there a statistical analysis done to determine this, or are you arbitrarily deciding it is so?*

The term "average" is arbitrary in this context, and was used here since the average of the May streamflow reconstructed using the tree-ring over this decade (1950-1960) is close from the overall May streamflow average value (1881-1980). We changed this descriptive term in the manuscript:

*Another significant difference exists over the 1950-1960 period, seen as an common decade by the tree ring reconstruction (reconstructed spring flood ranging from 47 to 87 [mm/m]), while being seen as a highly variable hydrological decade for the ANATEM reconstruction, with high values for the first five years (around 110 [mm/m] for the 1950-1955 period), and then two very low values (around 20 [mm/m] for the 1956-1957 period), finally followed by three high value years (around 110 [mm/m] for the 1958-1960 period).*

*37. Section 5: I would like to see a discussion of the parameters and limitations of the rainfall-runoff model. Were assumptions made with the rainfall-runoff model reasonable for this application?*

We added a discussion about the rainfall-runoff transformation in this section, arguing that the assumptions made are reasonable regarding the performances obtained by the rainfall-runoff model over the calibration period (presented in the results section):

*Finally, the reconstructed climatic time series are transformed into streamflow time series thanks to a daily rainfall-runoff model, previously calibrated over the relatively short observation period (with really good calibration performances). The use of one model, one objective function and one parameter set is questionable. Quantifying the sensitivity of the obtained reconstruction to the hydrological modeling assumptions made was out of the scope of this paper, but definitively deserves further research, especially considering the issue of uncertainty due to rainfall-runoff model parameters in a changing climate*

*38. Line 779: change "representing" to "represent"*

Agreed.

*39. Lines 799-812: I do not follow the text here. What limited performances are being referred to? What did Kuentz et al. (2015) highlight? How does the work have a perspective of finding an additional series? Is that done and described (I don't think so, but I couldn't really tell what was being stated here)? Please elaborate more on how variables like relative humidity, precipitable water content (what is this?), and local pressure measurements would be used. Would they be used in the rainfall-runoff model? Would they be used to reconstruct precipitation or air temperature? Where would these variables come from? Are they something that you can get from geopotential height? When reconstructing into the past, how you do you estimate these variables? Or are you intending to just reconstruct back through the observational record rather than for centuries as would be done with paleoreconstructions using tree-ring data?*

The limited performances referred here are the inability of the ANA approach to reproduce the long-term trend of climatic series (here temperature and precipitation), as already pointed out by Kuentz *et al.* (2015). Unfortunately, none long precipitation and temperature series are available in the studied region. The perspectives are thus to improve the current methodology and particularly testing variables available through the reanalysis for the analogy. Several authors used variables such as air temperature, vertical velocity and humidity at different atmospheric levels (variables produced by the 20CR reanalysis and thus available from 1851 to 2011) to find analogue dates and finally reconstruct daily air temperature and precipitation series. Trying to use such variables for the reconstruction and compare the obtained performances with and without these additional variables is an interesting perspective:

*The inability of the analogue approach to reproduce the interannual precipitation variability - already highlighted by Kuentz et al. (2015) over 22 French catchments – is due to the absence of a local reference climatic time series, unlike for temperature reconstruction, where a local temperature time series is used, and ensures that the simulated interannual temperature variability is reproduced efficiently. Finding an additional series which significantly improves the precipitation reconstruction is a major perspective of this work. The use of variables produced by the available reanalyses (e.g., relative humidity, precipitable water content) for finding analogue dates will be investigated, along with the testing of time series of local pressure measurements. For example, Caillouet et al. (2016) showed that adding the sea surface temperature variable to the temperature, geopotential, vertical velocity and humidity for finding analogue dates significantly improves the reconstruction of air temperature and precipitation over France.*

*40. Lines 813-823: Although the sensitivity analyses results are not shown, it would be useful to know what variables or approaches were sensitive. I did not follow the last sentence – was this lack of uncertainty shown in the results, and if so, can the authors point the reader to what they are referring to?*

*(This comment is found also in the general comment of the Referee #2).*

Several results of this sensitivity analysis (e.g. the spatial domain considered for the analogy) are now presented in a new Appendix part added to the manuscript (Appendix A). The last sentence was: "*Interestingly, the uncertainty due to the use of five members of the 20CR reanalysis appears to be limited, and even null from 1940 onward*". Yes, this "lack of uncertainty" is shown in results, see for example the Figure 8 (now Figure 9): it is impossible to distinguish the 5 ANATEM average series after 1940, highlighting that considering 5 different members of the 20CR reanalysis has a negligible impact on the reconstruction of the mean annual streamflow. We now explicitly point the reader to this figure in the manuscript:

*Interestingly, the uncertainty due to the use of five members of the 20CR reanalysis appears to be limited, and even null from 1940 onward. See for example Figure 9 which presents the centennial ANATEM streamflow reconstructions: it is impossible to distinguish the five ANATEM average series after 1940, highlighting that considering five different members of the 20CR reanalysis as inputs of the reconstruction method has a negligible impact on the reconstruction of the mean annual streamflow.*

*41. Line 825: Should "model" be added after "rainfall-runoff"?*

Yes, we added "model" after "rainfall-runoff".

*42. Lines 824-839: I do not follow what this paragraph is arguing. How (and why) would the parameter set change in changing climate? What parameter set are you talking about – the ones for the rainfall-runoff model, or perhaps the ones for Equation (1)? Please reword the entire paragraph to be more clear.*

This paragraph is intend to reminding and discussing the assumptions made when using a (calibrated) rainfall-runoff model over a climatically-contrasted and long period of time. We thus talk about the parameter set of the rainfall-runoff model, obtained after a calibration over a short period (here 17 years). Numerous authors thus highlighted that calibrated parameter sets are dependent on the climate of the calibration period and that the rainfall-runoff models show limited performances when applied over periods that are climatically contrasted regarding to the climate of the calibration period. It is clearly out of the scope of this paper to quantify the sensitivity of the streamflow reconstruction to these "stationary" assumptions, but it is an interesting perspective of this work. We reworded this paragraph to be clear:

*Finally, the reconstructed climatic time series are transformed into streamflow time series thanks to a daily rainfall-runoff model, previously calibrated over the relatively short observation period. The use of one model, one objective function and one parameter set is questionable. Quantifying the sensitivity of the obtained reconstruction to the hydrological modeling assumptions made was out of the scope of this paper, but definitively deserves further research, especially considering the issue of uncertainty due to rainfall-runoff model parameters in a changing climate. Thus, numerous authors highlighted that calibrated parameters of rainfall-runoff models are dependent on the climate of the calibration period and that performance decreases when applied over periods where the climate differs from that of calibration period (e.g., Merz et al. 2011; Coron et al. 2012 and Brigode et al. 2013b). Thus, testing different calibration strategies (e.g., bootstrap calibration used by Brigode et al. 2015), testing particular objective functions especially devoted to the final study objective (e.g., studying mean annual streamflow), and adapting the time step of the rainfall-runoff model to the objective would be interesting for future works.*

*43. Line 859: change "focusing" to "focus"*

Agreed.

*44. Figures 3, 4, 5, 6, 8, 9: I have a very difficult time making out the 5*20 ANATEM or 5*20 ANA data in these figures. I cannot distinguish 5*ANA from 5*20 ANATEM in Figure 3. I suggest the authors consider using some different colors for these lines or symbols.*

We agree that several lines are impossible to see or to distinguish on these figures. Even if this is a significant and interesting result (meaning that there is no dispersion between simulations or no difference between the observation and the simulation), we changed the colors between ANA and ANATEM (in Figures 5, 6 and 7) in order to distinguish the different simulations.

*45. Figure 7: Suggest moving "(a)" before "mean annual streamflow" and "(b)" before "May monthly"*

Agreed.

*46. Figure 9: Should the reference to Nicault et al. (2014) in the caption actually be to Boucher et al. (2011)?*

Yes, we corrected this mistake.

**2    REFEREE #2**

**2.1    General comments**

*The paper is well written and is in a form very similar to other paper on the paleoclimate text. It is rather long, but using several methods, this is necessary to present everything. Nevertheless, there is not always justification of the choices. For instance the choice of the zone used for the geopotential is not justified. And some parameters for the different models are not explicit. If possible it would be nice to integrate them in a way, but I know it is an issue because the paper will be longer. Because it is a long paper using several concepts, I would recommend to the author to summarize in a flow-chart figure each step of their methodology to reach streamflow. It would make it easier for the reader to follow the whole text. If the author can take this remarks into account, the paper will be nearly ready for publication.*

The tests performed for choosing the spatial domain considered for the geopotential height field are now presented in a new Appendix part of the manuscript (Appendix A).

We also added several paragraphs in order to fully describe the rainfall-runoff model and its snowmelt routine and how are calibrated the parameters (cf. answers to Referee #1 general comments).

Moreover, wed added a flowchart summarizing the reconstruction methodology applied (Figure 3).

**2.2    Specific comments**

*Fig 1: I do not recognize the catchment on figure 1b? why?*

The studied catchment is one of the 211 cQ2 catchments is thus plotted in the Figure 1b, but was hidden by an intermediate sub-catchment. The Caniapiscau catchment is now highlighted in the Figure 1b with shading lines.

*Page 3 line 4 add reference after "dendrohydrology".*

We added the reference to the review of Loaiciga *et al.* (1993).

*Legend figure 4: add "for" 1950??*

Agreed.

*Page 15 line 1: blank after the dot.*

We added a space.

*Figure 9: I do not understand tree ring reference to Nicault and Boucher in b?*

The "tree-ring series" presented in the Figure 9 is from Boucher *et al.* (2011), we thus corrected the Figure 9 (now Figure 10) legend.

**3    CITED REFERENCES**

[revised manuscript text omitted]

**1. Calibration of the R-R model**

Structure of the CemaNeigeGR4J daily R-R model:

[Figure]

Calibration of the 6 parameters ($X1,…, X6$) over the calibration period, with the KGE objective function.

**2. Finding analogue dates (ANA)**

Analysis of a 5-member ensemble (M1 to M5) of daily geopotential height fields (here at 500 hPa and 1000 hPa) over a given region:

[Figure]

Calculation of $D_{TW}$ distances for finding analogues dates in the observation period:

$$D_{TW} = D_{TW(Z500@0h)} + D_{TW(Z500@24h)} + D_{TW(Z1000@0h)} + D_{TW(Z1000@24h)}$$

For each day and each member M, selection of 20 analogues dates:

| | | | | | |
|---|---|---|---|---|---|
| M1 | 1984-01-23 | 1959-02-13 | | 1963-11-20 | 2007-12-18 |
| | 1991-12-12 | 1961-01-11 | | 1957-02-06 | 1989-11-05 |
| | – | – | – | – | – |
| | 1988-01-16 | 1953-12-25 | | 1975-01-06 | 2007-12-19 |
| M2 | 1984-01-23 | 1974-12-27 | | 1963-11-20 | 1979-11-19 |
| | 1990-11-30 | 1961-01-11 | | 1957-02-06 | 1971-11-13 |
| | – | – | – | – | – |
| | 1957-02-02 | 1990-02-19 | | 1988-01-29 | 1976-12-04 |
| M5 | 1984-01-23 | 1961-01-11 | | 1963-11-20 | 2007-12-18 |
| | 1989-01-13 | 1962-01-25 | | 1957-02-06 | 1971-11-13 |
| | – | – | – | – | – |
| | 1993-11-09 | 1965-11-04 | | 1962-01-14 | 1958-11-16 |

**3. Reconstruction of a daily climatic (P and T) ensemble (ANATEM)**

Resampling of observed climatic data using analogue dates (ANA) and correction of the obtained ensemble with a local regression model if long-term reference series are available (ANATEM, here applied for T):

[Figure]

[Figure]

**4. Reconstruction of a daily streamflow ensemble**

Transformation of the climatic ensemble into a streamflow ensemble using the CemaNeigeGR4J ($X1,…,X6$) parameters:

[Figure]

[Figure]

*Figure 3 : Illustration of the four-step methodology used for the reconstruction of a daily streamflow ensemble (R-R stands for rainfall-runoff, E for potential evapotranspiration, T for air temperature, P for precipitation, Q for streamflow).*

**3.1 Step 1: calibration of the rainfall-runoff model**

The GR4J (Perrin et al., 2003) rainfall-runoff model was used to transform the climatic ensemble into an ensemble of streamflow time series. GR4J is an efficient and parsimonious (only four free parameters to be calibrated) daily lumped and continuous model, which, when it is combined with its snow accumulation and melt  routine, CemaNeige (Valéry et al., 2014), is well suited for the hydrological modeling of snow-dominated catchments. GR4J and CemaNeige (model-pair hereafter denoted as CemaNeigeGR4J) were recently evaluated over several catchments located in Quebec (e.g., Seiller et al., 2012; Valéry et al., 2014) and showed good modelling performances.

The structure of the CemaNeigeGR4J model is presented in the Figure 3. GR4J is based on two non-linear stores (production and routing stores) and a unit-hydrograph, while CemaNeige is a degree-day snow accounting routine, which divides the studied catchment into five elevation bands. CemaNeigeGR4J uses as inputs daily series of precipitation, minimal and maximal air temperatures and a daily potential evapotranspiration series, calculated using Oudin et al. (2005) formula, designed for rainfall-runoff modelling. CemaNeigeGR4J produces daily streamflow series.

GR4J and CemaNeige have 4 and 2 free parameters to calibrate, respectively. These 6 parameters - highlighted in Figure 3 and described in Table 1 -  were calibrated  conjointly over the same calibration period  using a local gradient search procedure, applied in combination with pre-screening of the parameter space (Perrin et al., 2008). The Kling and Gupta Efficiency criterion (Gupta et al., 2009, hereafter denoted as KGE) was used as objective function. The KGE criterion ranges between -∞ and 1 (perfect simulation) and is calculated as follows:

$$\text{KGE} = 1 - \sqrt{(\beta - 1)^2 + (\alpha - 1)^2 + (r - 1)^2} \tag{1}$$

With:

[revised manuscript text omitted]

10.1002/2014JD022635

Compo GP, Whitaker JS, Sardeshmukh PD, et al (2011) The Twentieth Century Reanalysis Project. Q J R Meteorol
Soc 137:1–28. doi: 10.1002/qj.776

Coron L, Andréassian V, Perrin C, et al (2012) Crash testing hydrological models in contrasted climate conditions:
an experiment on 216 Australian catchments. Water Resour Res 48:W05552. doi:
10.1029/2011WR011721

Cowtan K, Way RG (2014) Coverage bias in the HadCRUT4 temperature series and its impact on recent
temperature trends. Q J R Meteorol Soc 140:1935–1944. doi: 10.1002/qj.2297

Crooks SM, Kay AL (2015) Simulation of river flow in the Thames over 120 years: Evidence of change in rainfall-
runoff response? J Hydrol Reg Stud 4, Part B:172–195. doi: 10.1016/j.ejrh.2015.05.014

Demarée GR, Ogilvie AEJ (2008) The Moravian missionaries at the Labrador coast and their centuries-long
contribution to instrumental meteorological observations. Clim Change 91:423–450. doi: 10.1007/s10584-
008-9420-2

Garçon R (1999) Modèle global pluie-débit pour la prévision et la prédétermination des crues. Houille Blanche 88–
95. doi: 10.1051/lhb/1999088

George SS, Nielsen E (2003) Palaeoflood records for the Red River, Manitoba, Canada, derived from anatomical
tree-ring signatures. The Holocene 13:547–555. doi: 10.1191/0959683603hl645rp

Gray ST, McCabe GJ (2010) A combined water balance and tree ring approach to understanding the potential
hydrologic effects of climate change in the central Rocky Mountain region. Water Resour Res 46:W05513.
doi: 10.1029/2008WR007650

GRDC (2015) Global Runoff Data Centre. Federal Institute of Hydrology (BfG), D-56002, Koblenz, Germany

Guay C, Minville M, Braun M (2015) A global portrait of hydrological changes at the 2050 horizon for the province
of Québec. Can Water Resour J Rev Can Ressour Hydr 40:285–302. doi:
10.1080/07011784.2015.1043583

Gupta HV, Kling H, Yilmaz KK, Martinez GF (2009) Decomposition of the mean squared error and NSE
performance criteria: Implications for improving hydrological modelling. J Hydrol 377:80–91. doi:
16/j.jhydrol.2009.08.003

Hernández-Henríquez MA, Mlynowski TJ, Déry SJ (2010) Reconstructing the Natural Streamflow of a Regulated
River: A Case Study of La Grande Rivière, Québec, Canada. Can Water Resour J Rev Can Ressour Hydr
35:301–316. doi: 10.4296/cwrj3503301

Hirsch RM (1982) A comparison of four streamflow record extension techniques. Water Resour Res 18:1081–1088.
doi: 10.1029/WR018i004p01081

Horton P, Jaboyedoff M, Metzger R, et al (2012) Spatial relationship between the atmospheric circulation and the
precipitation measured in the western Swiss Alps by means of the analogue method. Nat Hazards Earth
Syst Sci 12:777–784. doi: 10.5194/nhess-12-777-2012

Huang J-G, Bergeron Y, Berninger F, et al (2013) Impact of Future Climate on Radial Growth of Four Major Boreal
Tree Species in the Eastern Canadian Boreal Forest. PLoS ONE 8:e56758. doi:
10.1371/journal.pone.0056758

Jandhyala VK, Liu P, Fotopoulos SB (2009) River stream flows in the northern Québec Labrador region: A
multivariate change point analysis via maximum likelihood. Water Resour Res 45:W02408. doi:
10.1029/2007WR006499

Jarvis A, Reuter HI, Nelson A, Guevara E (2008) Hole-filled SRTM for the globe Version 4.

Kuentz A, Mathevet T, Gailhard J, et al (2013) Over 100 years of climatic and hydrologic variability of a
Mediterranean and mountainous watershed: the Durance River. In: Cold and Mountain Region
Hydrological Systems Under Climate Change: Towards Improved Projections. IAHS Publications,
Gothenburg, Sweden, pp 19–25

Kuentz A, Mathevet T, Gailhard J, Hingray B (2015) Building long-term and high spatio-temporal resolution
precipitation and air temperature reanalyses by mixing local observations and global atmospheric
reanalyses: the ANATEM model. Hydrol Earth Syst Sci 19:2717–2736. doi: 10.5194/hess-19-2717-2015

Loaiciga HA, Haston L, Michaelsen J (1993) Dendrohydrology and long-term hydrologic phenomena. Rev Geophys
31:151–171.

Mekis É, Vincent LA (2011) An Overview of the Second Generation Adjusted Daily Precipitation Dataset for Trend
Analysis in Canada. Atmosphere-Ocean 49:163–177. doi: 10.1080/07055900.2011.583910

Meko DM, Woodhouse CA (2011) Application of Streamflow Reconstruction to Water Resources Management. In:
Hughes MK, Swetnam TW, Diaz HF (eds) Dendroclimatology. Springer Netherlands, pp 231–261

Merz R, Parajka J, Blöschl G (2011) Time stability of catchment model parameters: Implications for climate impact
analyses. Water Resour Res. doi: 10.1029/2010WR009505

Montanari A (2012) Hydrology of the Po River: looking for changing patterns in river discharge. Hydrol Earth Syst
Sci 16:3739–3747. doi: 10.5194/hess-16-3739-2012

Nicault A, Boucher E, Bégin C, et al (2014) Hydrological reconstruction from tree-ring multi-proxies over the last
two centuries at the Caniapiscau Reservoir, northern Québec, Canada. J Hydrol 513:435–445. doi:
10.1016/j.jhydrol.2014.03.054

Obled C, Bontron G, Garçon R (2002) Quantitative precipitation forecasts: a statistical adaptation of model outputs
through an analogues sorting approach. Atmospheric Res 63:303–324. doi: 10.1016/S0169-
8095(02)00038-8

Oudin L, Hervieu F, Michel C, et al (2005) Which potential evapotranspiration input for a lumped rainfall–runoff
model?: Part 2—Towards a simple and efficient potential evapotranspiration model for rainfall–runoff
modelling. J Hydrol 303:290–306. doi: 10.1016/j.jhydrol.2004.08.026

Patskoski J, Sankarasubramanian A, Wang H (2015) Reconstructed streamflow using SST and tree-ring
chronologies over the southeastern United States. J Hydrol 527:761–775. doi:
10.1016/j.jhydrol.2015.05.041

Perreault L, Garçon R, Gaudet J (2007) Analyse de séquences de variables aléatoires hydrologiques à l'aide de
modèles de changement de régime exploitant des variables atmosphériques. Houille Blanche 111–123.
doi: 10.1051/lhb:2007091

Perreault L, Parent É, Bernier J, et al (2000) Retrospective multivariate Bayesian change-point analysis: A
simultaneous single change in the mean of several hydrological sequences. Stoch Environ Res Risk
Assess 14:243–261. doi: 10.1007/s004770000051

Perrin C, Andréassian V, Rojas Serna C, et al (2008) Discrete parameterization of hydrological models: Evaluating
the use of parameter sets libraries over 900 catchments. Water Resour Res 44:W08447. doi:
10.1029/2007WR006579

Perrin C, Michel C, Andréassian V (2003) Improvement of a parsimonious model for streamflow simulation. J Hydrol
279:275–289. doi: 10.1016/S0022-1694(03)00225-7

R Core Team (2014) R: A Language and Environment for Statistical Computing. R Foundation for Statistical
Computing, Vienna, Austria

Radanovics S, Vidal J-P, Sauquet E, et al (2013) Optimising predictor domains for spatially coherent precipitation
downscaling. Hydrol Earth Syst Sci 17:4189–4208. doi: 10.5194/hess-17-4189-2013

Rohde R, Muller RA, Jacobsen R, et al (2013) A New Estimate of the Average Earth Surface Land Temperature
Spanning 1753 to 2011. Geoinformatics Geostat Overv 1:1–7. doi: 10.4172/2327-4581.1000101

Saito L, Biondi F, Devkota R, et al (2015) A water balance approach for reconstructing streamflow using tree-ring
proxy records. J Hydrol 529, Part 2:535–547. doi: 10.1016/j.jhydrol.2014.11.022

Seiller G, Anctil F, Perrin C (2012) Multimodel evaluation of twenty lumped hydrological models under contrasted
climate conditions. Hydrol Earth Syst Sci 16:1171–1189. doi: 10.5194/hess-16-1171-2012

Slonosky VC (2014) Daily minimum and maximum temperature in the St-Lawrence Valley, Quebec: two centuries
of climatic observations from Canada. Int J Climatol. doi: 10.1002/joc.4085

Subedi N, Sharma M (2013) Climate-diameter growth relationships of black spruce and jack pine trees in boreal
Ontario, Canada. Glob Change Biol 19:505–516. doi: 10.1111/gcb.12033

Tapsoba D, Fortin V, Anctil F, Haché M (2005) Apport de la technique du krigeage avec dérive externe pour une
cartographie raisonnée de l'équivalent en eau de la neige : Application aux bassins de la rivière Gatineau.
Can J Civ Eng 32:289–297. doi: 10.1139/l04-110

Teng J, Potter NJ, Chiew FHS, et al (2015) How does bias correction of regional climate model precipitation affect
modelled runoff? Hydrol Earth Syst Sci 19:711–728. doi: 10.5194/hess-19-711-2015

Teutschbein C, Seibert J (2013) Is bias correction of regional climate model (RCM) simulations possible for non-
stationary conditions? Hydrol Earth Syst Sci 17:5061–5077. doi: 10.5194/hess-17-5061-2013

Teweles J, Wobus H (1954) Verification of prognosis charts. Bull Am Meteorol Soc 35:455–463.

Thorndycraft VR, Benito G, Rico M, et al (2005) A long-term flood discharge record derived from slackwater flood
deposits of the Llobregat River, NE Spain. J Hydrol 313:16–31. doi: 10.1016/j.jhydrol.2005.02.003

Valéry A, Andréassian V, Perrin C (2014) "As simple as possible but not simpler": what is useful in a temperature-
based snow-accounting routine? Part 2 - Sensitivity analysis of the Cemaneige snow accounting routine
on 380 catchments. J Hydrol. doi: 10.1016/j.jhydrol.2014.04.058

Velázquez JA, Troin M, Caya D, Brissette F (2015) Evaluating the Time-Invariance Hypothesis of Climate Model
Bias Correction: Implications for Hydrological Impact Studies. J Hydrometeorol 16:2013–2026. doi:
10.1175/JHM-D-14-0159.1

Vincent LA, Wang XL, Milewska EJ, et al (2012) A second generation of homogenized Canadian monthly surface
air temperature for climate trend analysis. J Geophys Res Atmospheres 117:D18110. doi:
10.1029/2012JD017859

Way RG, Viau AE (2014) Natural and forced air temperature variability in the Labrador region of Canada during the
past century. Theor Appl Climatol 121:413–424. doi: 10.1007/s00704-014-1248-2

Wetterhall F, Halldin S, Xu C (2005) Statistical precipitation downscaling in central Sweden with the analogue
method. J Hydrol 306:174–190. doi: 10.1016/j.jhydrol.2004.09.008

Wilson CV (1988) The summer season along the east coast of Hudson Bay during the nineteenth century. Part III:
Summer thermal and wetness indices B. The indices, 1800-1900. Environment Canada, Downsview,
Ontario, Canada

---

## Author Response (AR2)

*Streamflow variability over 1881–2011 period in northern Quebec:*
*Comparison of hydrological reconstructions based on tree rings and on*
*geopotential height field reanalysis*
**Paper *cp-2016-5**

by Brigode, P.; Brissette, F.; Nicault, A.; Perreault, L.; Kuentz, A.; Mathevet, T. & Gailhard, J.

**Point-by-point reply to the editor comments and markep-up manuscript version**

Comments and suggestions made by the editor are gratefully acknowledged. We modified the text in response to the main criticisms. In the following, we list the editor comments (in *italic and blue*), we provide specific responses to these comments (in black) and finally we present a marked-up manuscript version.

**1  EDITOR COMMENTS**

*1. You may way want to cite the previous paper of mine (Schenk and Zorita, Climate of the Past doi:10.5194/cp-8-1681-2012) as it is directly related to the reconstruction method you are using in this manuscript.*

This Climate of the Past paper being highly relevant regarding our reconstruction methodology, we now cited this work in the new version and thank the editor for this suggestion:

*The ANATEM method (Kuentz et al., 2015) is built on the combination of two approaches: (i) the ANA (which stands for "ANAlogue") approach, that aims to find, for a given day, a given number of analogue days, based on the similarity of synoptic circulation (Obled et al., 2002, Schenk and Zorita, 2012) and (ii) the TEM (which stands for "TEMoin", the French word for "witness") approach, which is a basic regression model that uses a continuous and long-term reference (the witness) climatic series to reconstruct past climate.*

*2. Page 2 line 10. There seems to be something missing in this sentence "and highlighting significant natural variability at the decadal Montanari (2012) stated."*

The editor is right, we thus corrected this sentence:

*For example, after studying a 90-year long daily streamflow series of the Po River (Italy), and highlighting significant natural variability at the decadal scale, Montanari (2012) stated that "more research efforts are needed to improve the interpretation of such long-term fluctuations".*

*3. Page 4, line 29.*

*I am aware that one of the previous reviewers requested a more detailed explanation of the concept of geopotential height, but I disagree with the reviewer on this point. I do not think it is necessary and that it distorts the overall text flow, but I leave it to your decision to keep or delete this paragraph.*

*'For example, if a station reports that the 500 hPa height at its location is 5600 meters, it means that the level of the atmosphere over that station at which the atmospheric pressure is 500 hPa is 5600 meters above sea level (example from the NOAA's National Weather Service).'*

We decided to delete this sentence:

*The similarity is based on geopotential height fields over a given spatial domain. A geopotential height is the height above sea level of a given pressure level. Note that for pressure levels close to sea level (typically 1000 hPa), the geopotential height can sometimes be negative.*

*4 Page 5, line 11*

*' see section 3.1.1 for more details'*

*section 3.1.1 does not exist. I think it should read 3.1 or maybe 3.3*

We corrected this mistake by referring to the section 3.3 in the new version:

*Of the 56 ensemble members constituting the 20CR reanalysis, the members 1 to 5 were extracted and used over this region (see section 3.3 for more details).*

*5. Page 15, line 37*

*'Averaging each ensemble of the considered 20CR members (blue points) results in better temporal correlations at the daily and yearly resolutions, but at the expense of lower variability reproduction performance.'*

*The last part of the sentence is unclear. I think you mean 'at the expense of a too small reconstructed variability '*

Yes, we modified this sentence according to the editor proposition:

*Averaging each ensemble of the considered 20CR members (blue points) results in better temporal correlations at the daily and yearly resolutions, but at the expense of a too small reconstructed variability.*

*6. Section 4.3.2 Correlation Centennial mean annual flow reconstructions (1881-2011)*

*I think it would be interesting to include in the description of the results the correlation - or other measure of similarity - between the observed run-off and the tree-ring reconstructed run-off*

We estimated the performances of the dendrohydrological reconstructions, added them in Figure 8 and thus compared the performances of the different streamflow reconstructions:

*The performances of the dendrohydrological reconstructions are also evaluated and are shown in Figure 8, highlighting that dendrohydrological reconstructions perform slightly better than ANATEM ones for the mean annual streamflow values while ANATEM reconstructions perform better than dendrohydrological ones for the May monthly flow values.*

*7. The caption of figure 10 is not correct. It is a repetition of caption 9. Caption 10 should refer to the spring run-off reconstruction and the comparison with Boucher (2015).*

We changed the figure caption:

[revised manuscript text omitted]

**1. Calibration of the R-R model**

Structure of the CemaNeigeGR4J daily R-R model:

[Figure]

**2. Finding analogue dates (ANA)**

Analysis of a 5-member ensemble (M1 to M5) of daily geopotential height fields (here at 500 hPa and 1000 hPa) over a given region:

[Figure]

Calibration of the 6 parameters (**X1**,..., **X6**) over the calibration period, with the KGE objective function.

Calculation of $D_{TW}$ distances for finding analogues dates in the observation period:

$$D_{TW} = D_{TW(Z500@0h)} + D_{TW(Z500@24h)} + D_{TW(Z1000@0h)} + D_{TW(Z1000@24h)}$$

For each day and each member M, selection of 20 analogues dates:

| | | | | | | | |
|---|---|---|---|---|---|---|---|
| M1 | 1984-01-23 | 1959-02-13 | | 1963-11-20 | 2007-12-18 |
| | 1991-12-12 | 1961-01-11 | | 1957-02-06 | 1989-11-05 |
| | – | – | – | – | – |
| | 1988-01-16 | 1953-12-25 | | 1975-01-06 | 2007-12-19 |
| M2 | 1984-01-23 | 1974-12-27 | | 1963-11-20 | 1979-11-19 |
| | 1990-11-30 | 1961-01-11 | | 1957-02-06 | 1971-11-13 |
| | – | – | – | – | – |
| | 1957-02-02 | 1990-02-19 | | 1988-01-29 | 1976-12-04 |
| M5 | 1984-01-23 | 1961-01-11 | | 1963-11-20 | 2007-12-18 |
| | 1989-01-13 | 1962-01-25 | | 1957-02-06 | 1971-11-13 |
| | – | – | – | – | – |
| | 1993-11-09 | 1965-11-04 | | 1962-01-14 | 1958-11-16 |

**3. Reconstruction of a daily climatic (P and T) ensemble (ANATEM)**

Resampling of observed climatic data using analogue dates (ANA) and correction of the obtained ensemble with a local regression model if long-term reference series are available (ANATEM, here applied for T):

[Figure]

[Figure]

**4. Reconstruction of a daily streamflow ensemble**

Transformation of the climatic ensemble into a streamflow ensemble using the CemaNeigeGR4J (**X1**,...,**X6**) parameters:

[revised manuscript text omitted]

10.1002/2014JD022635

Compo GP, Whitaker JS, Sardeshmukh PD, et al (2011) The Twentieth Century Reanalysis Project. Q J R Meteorol
Soc 137:1–28. doi: 10.1002/qj.776

Coron L, Andréassian V, Perrin C, et al (2012) Crash testing hydrological models in contrasted climate conditions:
an experiment on 216 Australian catchments. Water Resour Res 48:W05552. doi:
10.1029/2011WR011721

Cowtan K, Way RG (2014) Coverage bias in the HadCRUT4 temperature series and its impact on recent
temperature trends. Q J R Meteorol Soc 140:1935–1944. doi: 10.1002/qj.2297

Crooks SM, Kay AL (2015) Simulation of river flow in the Thames over 120 years: Evidence of change in rainfall-
runoff response? J Hydrol Reg Stud 4, Part B:172–195. doi: 10.1016/j.ejrh.2015.05.014

Demarée GR, Ogilvie AEJ (2008) The Moravian missionaries at the Labrador coast and their centuries-long
contribution to instrumental meteorological observations. Clim Change 91:423–450. doi: 10.1007/s10584-
008-9420-2

George SS, Nielsen E (2003) Palaeoflood records for the Red River, Manitoba, Canada, derived from anatomical
tree-ring signatures. The Holocene 13:547–555. doi: 10.1191/0959683603hl645rp

Gray ST, McCabe GJ (2010) A combined water balance and tree ring approach to understanding the potential
hydrologic effects of climate change in the central Rocky Mountain region. Water Resour Res 46:W05513.
doi: 10.1029/2008WR007650

GRDC (2015) Global Runoff Data Centre. Federal Institute of Hydrology (BfG), D-56002, Koblenz, Germany

Guay C, Minville M, Braun M (2015) A global portrait of hydrological changes at the 2050 horizon for the province
of Québec. Can Water Resour J Rev Can Ressour Hydr 40:285–302. doi:
10.1080/07011784.2015.1043583

Gupta HV, Kling H, Yilmaz KK, Martinez GF (2009) Decomposition of the mean squared error and NSE
performance criteria: Implications for improving hydrological modelling. J Hydrol 377:80–91. doi:
16/j.jhydrol.2009.08.003

Hernández-Henríquez MA, Mlynowski TJ, Déry SJ (2010) Reconstructing the Natural Streamflow of a Regulated
River: A Case Study of La Grande Rivière, Québec, Canada. Can Water Resour J Rev Can Ressour Hydr
35:301–316. doi: 10.4296/cwrj3503301

Hirsch RM (1982) A comparison of four streamflow record extension techniques. Water Resour Res 18:1081–1088.
doi: 10.1029/WR018i004p01081

Horton P, Jaboyedoff M, Metzger R, et al (2012) Spatial relationship between the atmospheric circulation and the
precipitation measured in the western Swiss Alps by means of the analogue method. Nat Hazards Earth
Syst Sci 12:777–784. doi: 10.5194/nhess-12-777-2012

Huang J-G, Bergeron Y, Berninger F, et al (2013) Impact of Future Climate on Radial Growth of Four Major Boreal
Tree Species in the Eastern Canadian Boreal Forest. PLoS ONE 8:e56758. doi:
10.1371/journal.pone.0056758

Jandhyala VK, Liu P, Fotopoulos SB (2009) River stream flows in the northern Québec Labrador region: A
multivariate change point analysis via maximum likelihood. Water Resour Res 45:W02408. doi:
10.1029/2007WR006499

Jarvis A, Reuter HI, Nelson A, Guevara E (2008) Hole-filled SRTM for the globe Version 4.

Kuentz A, Mathevet T, Gailhard J, et al (2013) Over 100 years of climatic and hydrologic variability of a
Mediterranean and mountainous watershed: the Durance River. In: Cold and Mountain Region
Hydrological Systems Under Climate Change: Towards Improved Projections. IAHS Publications,
Gothenburg, Sweden, pp 19–25

Kuentz A, Mathevet T, Gailhard J, Hingray B (2015) Building long-term and high spatio-temporal resolution
precipitation and air temperature reanalyses by mixing local observations and global atmospheric
reanalyses: the ANATEM model. Hydrol Earth Syst Sci 19:2717–2736. doi: 10.5194/hess-19-2717-2015

Loaiciga HA, Haston L, Michaelsen J (1993) Dendrohydrology and long-term hydrologic phenomena. Rev Geophys
31:151–171.

Mekis É, Vincent LA (2011) An Overview of the Second Generation Adjusted Daily Precipitation Dataset for Trend
Analysis in Canada. Atmosphere-Ocean 49:163–177. doi: 10.1080/07055900.2011.583910

Meko DM, Woodhouse CA (2011) Application of Streamflow Reconstruction to Water Resources Management. In:
Hughes MK, Swetnam TW, Diaz HF (eds) Dendroclimatology. Springer Netherlands, pp 231–261

Merz R, Parajka J, Blöschl G (2011) Time stability of catchment model parameters: Implications for climate impact
analyses. Water Resour Res. doi: 10.1029/2010WR009505

Montanari A (2012) Hydrology of the Po River: looking for changing patterns in river discharge. Hydrol Earth Syst Sci 16:3739–3747. doi: 10.5194/hess-16-3739-2012

Nicault A, Boucher E, Bégin C, et al (2014) Hydrological reconstruction from tree-ring multi-proxies over the last two centuries at the Caniapiscau Reservoir, northern Québec, Canada. J Hydrol 513:435–445. doi: 10.1016/j.jhydrol.2014.03.054

Obled C, Bontron G, Garçon R (2002) Quantitative precipitation forecasts: a statistical adaptation of model outputs through an analogues sorting approach. Atmospheric Res 63:303–324. doi: 10.1016/S0169-8095(02)00038-8

Oudin L, Hervieu F, Michel C, et al (2005) Which potential evapotranspiration input for a lumped rainfall–runoff model?: Part 2—Towards a simple and efficient potential evapotranspiration model for rainfall–runoff modelling. J Hydrol 303:290–306. doi: 10.1016/j.jhydrol.2004.08.026

Patskoski J, Sankarasubramanian A, Wang H (2015) Reconstructed streamflow using SST and tree-ring chronologies over the southeastern United States. J Hydrol 527:761–775. doi: 10.1016/j.jhydrol.2015.05.041

Perreault L, Garçon R, Gaudet J (2007) Analyse de séquences de variables aléatoires hydrologiques à l'aide de modèles de changement de régime exploitant des variables atmosphériques. Houille Blanche 111–123. doi: 10.1051/lhb:2007091

Perreault L, Parent É, Bernier J, et al (2000) Retrospective multivariate Bayesian change-point analysis: A simultaneous single change in the mean of several hydrological sequences. Stoch Environ Res Risk Assess 14:243–261. doi: 10.1007/s004770000051

Perrin C, Andréassian V, Rojas Serna C, et al (2008) Discrete parameterization of hydrological models: Evaluating the use of parameter sets libraries over 900 catchments. Water Resour Res 44:W08447. doi: 10.1029/2007WR006579

Perrin C, Michel C, Andréassian V (2003) Improvement of a parsimonious model for streamflow simulation. J Hydrol 279:275–289. doi: 10.1016/S0022-1694(03)00225-7

R Core Team (2014) R: A Language and Environment for Statistical Computing. R Foundation for Statistical Computing, Vienna, Austria

Radanovics S, Vidal J-P, Sauquet E, et al (2013) Optimising predictor domains for spatially coherent precipitation downscaling. Hydrol Earth Syst Sci 17:4189–4208. doi: 10.5194/hess-17-4189-2013

Rohde R, Muller RA, Jacobsen R, et al (2013) A New Estimate of the Average Earth Surface Land Temperature Spanning 1753 to 2011. Geoinformatics Geostat Overv 1:1–7. doi: 10.4172/2327-4581.1000101

Saito L, Biondi F, Devkota R, et al (2015) A water balance approach for reconstructing streamflow using tree-ring proxy records. J Hydrol 529, Part 2:535–547. doi: 10.1016/j.jhydrol.2014.11.022

Schenk F, Zorita E (2012) Reconstruction of high resolution atmospheric fields for Northern Europe using analog-upscaling. Clim Past 8:1681–1703. doi: 10.5194/cp-8-1681-2012

Seiller G, Anctil F, Perrin C (2012) Multimodel evaluation of twenty lumped hydrological models under contrasted climate conditions. Hydrol Earth Syst Sci 16:1171–1189. doi: 10.5194/hess-16-1171-2012

Slonosky VC (2014) Daily minimum and maximum temperature in the St-Lawrence Valley, Quebec: two centuries of climatic observations from Canada. Int J Climatol. doi: 10.1002/joc.4085

Subedi N, Sharma M (2013) Climate-diameter growth relationships of black spruce and jack pine trees in boreal
  Ontario, Canada. Glob Change Biol 19:505–516. doi: 10.1111/gcb.12033

Tapsoba D, Fortin V, Anctil F, Haché M (2005) Apport de la technique du krigeage avec dérive externe pour une
  cartographie raisonnée de l'équivalent en eau de la neige : Application aux bassins de la rivière Gatineau.
  Can J Civ Eng 32:289–297. doi: 10.1139/l04-110

Teng J, Potter NJ, Chiew FHS, et al (2015) How does bias correction of regional climate model precipitation affect
  modelled runoff? Hydrol Earth Syst Sci 19:711–728. doi: 10.5194/hess-19-711-2015

Teutschbein C, Seibert J (2013) Is bias correction of regional climate model (RCM) simulations possible for non-
  stationary conditions? Hydrol Earth Syst Sci 17:5061–5077. doi: 10.5194/hess-17-5061-2013

Teweles J, Wobus H (1954) Verification of prognosis charts. Bull Am Meteorol Soc 35:455–463.

Thorndycraft VR, Benito G, Rico M, et al (2005) A long-term flood discharge record derived from slackwater flood
  deposits of the Llobregat River, NE Spain. J Hydrol 313:16–31. doi: 10.1016/j.jhydrol.2005.02.003

Valéry A, Andréassian V, Perrin C (2014) "As simple as possible but not simpler": what is useful in a temperature-
  based snow-accounting routine? Part 2 - Sensitivity analysis of the Cemaneige snow accounting routine
  on 380 catchments. J Hydrol. doi: 10.1016/j.jhydrol.2014.04.058

Velázquez JA, Troin M, Caya D, Brissette F (2015) Evaluating the Time-Invariance Hypothesis of Climate Model
  Bias Correction: Implications for Hydrological Impact Studies. J Hydrometeorol 16:2013–2026. doi:
  10.1175/JHM-D-14-0159.1

Vincent LA, Wang XL, Milewska EJ, et al (2012) A second generation of homogenized Canadian monthly surface
  air temperature for climate trend analysis. J Geophys Res Atmospheres 117:D18110. doi:
  10.1029/2012JD017859

Way RG, Viau AE (2014) Natural and forced air temperature variability in the Labrador region of Canada during the
  past century. Theor Appl Climatol 121:413–424. doi: 10.1007/s00704-014-1248-2

Wetterhall F, Halldin S, Xu C (2005) Statistical precipitation downscaling in central Sweden with the analogue
  method. J Hydrol 306:174–190. doi: 10.1016/j.jhydrol.2004.09.008

Wilson CV (1988) The summer season along the east coast of Hudson Bay during the nineteenth century. Part III:
  Summer thermal and wetness indices B. The indices, 1800-1900. Environment Canada, Downsview,
  Ontario, Canada